

# Methanesulfonic acid (MSA) migration in polar ice: Data synthesis and theory

Matthew Osman[1], Sarah B. Das[2], Olivier Marchal[2], Matthew J. Evans[3]

[1]Massachusetts Institute of Technology/Woods Hole Oceanographic Institution Joint Program in Oceanography/Applied Ocean Sciences and Engineering, Woods Hole Oceanographic Institution, Woods Hole, MA, 02543, USA
[2]Dept. of Geology and Geophysics, Woods Hole Oceanographic Institution, Woods Hole, MA, USA
[3]Dept. of Chemistry, Wheaton College, Wheaton, MA, USA

*Correspondence to*: Matthew Osman (osmanm@mit.edu)

**Abstract** Methanesulfonic acid (MSA; $CH_3SO_3H$) in polar ice is a unique proxy of marine primary productivity, synoptic atmospheric transport, and regional sea ice behavior. However, MSA can be mobile within the firn and ice matrix, a post-depositional process that is well known but poorly understood and documented, leading to uncertainties in the integrity of the MSA paleoclimatic signal.

Here, we use a compilation of 22 ice core MSA records from Greenland and Antarctica and a model of soluble impurity transport in order to comprehensively investigate the vertical migration of MSA from summer layers, where MSA is originally deposited, to adjacent winter layers in polar ice. The shallowest depths of MSA migration reported in our compilation vary over a wide range (~2 m to 400 m), and our analysis suggests that these depths are positively correlated with snow accumulation rate

and negatively correlated with ice concentration of $Na^+$ (typically the most abundant cationic sea salt).

Although the considered soluble impurity transport model provides a useful mechanistic framework for studying MSA migration, it remains limited by inadequate constraints on key physico-chemical parameters, most notably, the diffusion coefficient of MSA in cold ice ($D_{MS}$). We derive a simplified version of the model, which includes $D_{MS}$ as the sole parameter, in order to illuminate aspects

of the migration process. Using this model, we show that the progressive phase alignment of MSA and $Na^+$ concentration peaks observed along a high-resolution West Antarctic core is most consistent with $10^{-12}\,m^2\,s^{-1} < D_{MS} < 10^{-11}\,m^2\,s^{-1}$, one order of magnitude greater than $D_{MS}$ values previously estimated





from laboratory studies. More generally, our data synthesis and model results suggest that (i) MSA migration may be fairly ubiquitous, particularly at coastal and (or) high accumulation regions across Greenland and Antarctica, and (ii) can significantly change annual and multi-year MSA concentration averages. Thus, in most cases, caution should be exercised when interpreting polar ice core MSA
records, although records that have undergone severe migration could still be useful for inferring decadal and lower-frequency climate variability.

## 1. Introduction

Measurements of soluble impurity species in polar ice cores provide important high-resolution proxies of past climatic phenomena, including past changes in sea-ice extent, marine and terrestrial
productivity, volcanism, biomass burning, atmospheric cycling, and anthropogenic pollution (e.g., Legrand and Mayewski, 1997). A foundational premise, however, is that these species undergo negligible post-depositional redistribution in the ice column, an assumption unsupported by numerous ice core records from Greenland and Antarctica. Processes acting within the upper firn layer, including wind pumping, diffusion, volatility, sublimation, and melt (Wolff et al., 1996), can affect the stability of
chemical species soon after deposition and undermine the climatic interpretation of these records down-core (Wagnon et al. 1999, Weller et al. 2004). Deeper in the ice column, both observation (e.g., Barnes et al., 2003b) and theory (Nye, 1991; Rempel et al. 2001, 2002) indicate the potential for solid and liquid-state chemical migration, impacting the stability of chemical species much later after deposition. In this study, we focus on one species particularly susceptible to vertical migration in polar ice,
methanesulfonic acid, or MSA ($CH_3SO_3H$; Pasteur and Mulvaney, 2000).

The processes leading to the production, transport, and deposition of MSA onto an ice sheet are complex (e.g., Abram, 2013). The progenitor compound of MSA, dimethylsulfoniopropionate, or DMSP ($(CH_3)_2S^+CH_2CH_2COO^-$), is produced by certain marine algae as an osmotic regulator (Dickson and Kirst, 1986). Planktonic life cycle processes ultimately release DMSP to the water column,
whereupon the ensuing bacterial-mediated cleavage of the compound promotes the formation of dimethylsulfide, or DMS ($(CH_3)_2S$), a highly-insoluble gaseous compound (Yoch, 2002). Once freed to the atmosphere, DMS is rapidly photo-oxidized (Saltzman et al., 1983), branching to form either non-





sea salt sulfate ($nssSO_4^{-2}$), or, to a lesser extent, MSA. Unlike $nssSO_4^{-2}$, however, DMS production appears to be the exclusive source of MSA (Abram, 2013).

Early studies on DMS and DMSP production in the Southern Ocean (Curran and Jones, 2000) and in Arctic waters (Leck and Persson, 1996) reported the greatest fluxes of DMS near the marginal sea

ice zone at the onset of spring – summer decay (Turner et al., 1995; Curran et al., 1998). Since concentrations of atmospheric MSA rapidly decrease with increasing altitude and distance from their marine source (the mean atmospheric lifetime of MSA is estimated to about 7 days; Hezel et al. 2011), deposition of MSA at coastal ice sheet localities near-ubiquitously exhibits well-defined annual peaks during the late-spring to summer months. Conversely, the lack of wintertime MSA deposition may be

jointly attributed to (1) limited marine productivity during polar darkness, (2) increased wintertime sea ice extent and, accordingly, atmospheric transport distances, and (or) (3) diminished atmospheric $OH^-$ concentrations, the primary oxidant of airborne DMS (Jourdain and Legrand, 2001). The strong seasonality and unique marine source of MSA in ice cores have led to its predominant use as a high-resolution proxy for past sea-ice cover (e.g., Welch et al., 1993; Abram et al., 2013).

The first authors to report migration of MSA in polar ice are Mulvaney et al. (1992), using data from Dolleman Plateau, Antarctica Peninsula. In the shallow portions of the Dolleman Plateau core, concentrations of MSA exhibited well-defined summer maxima, as expected. With increasing depth in the ice column, however, a distinctive shift to predominantly winter [$MS^-$] maxima was found (hereafter, MSA is denoted as $MS^-$ when referring to its anionic form, $CH_3SO_3^-$, as measured by ion

chromatography). Concluding that a change in the seasonality of peak MSA production and (or) deposition was unlikely, the authors postulated this shift to result from post-depositional vertical migration. Since then, numerous ice core studies from both Antarctica and Greenland have reported the migration phenomenon over a wide range of depths, temperatures, and ionic concentrations in the ice column (Table 1 and references therein).

Despite the importance of [$MS^-$] records in paleoclimatic studies, there have been few systematic evaluations of the environmental and (or) chemical conditions promoting MSA migration from summer-to-winter layers in polar ice. Furthermore, aspects of this migration process exhibit behavior distinct from other post-depositional processes, and are important to understand mechanistically. First, the [$MS^-$]



maxima that are formed in winter are converse to what would be expected from typical (Fickian) diffusion, which would instead weaken the amplitude of summertime [MS⁻] peaks. Second, MSA movement has been reported to occur in the up-core direction (Curran et al., 2002), ruling out gravitational forcing as the sole mechanism for migration. These observations, corroborated by evidence

for highly concentrated regions of sulfuric acid at the interface of individual ice crystals (Mulvaney et al. 1988), led to speculation that liquid migration of soluble impurities could occur along the boundaries of individual ice crystals, likely along concentration gradients (Mulvaney et al. 1992). However, critical questions remain. For example, why should MSA in particular exhibit migration, while associated soluble impurities and acids (e.g., $nssSO_4^{2-}$) do not (Pasteur et al., 1999)? How could diffusion result in

clearly defined concentration maxima in winter layers? Could the "trapping" of migrating MSA in the adjacent winter layer stem any subsequent "spillover" of MSA into an adjacent annual layer, as assumed in prior studies (e.g., Kreutz et al. 1998; Pasteur and Mulvaney, 2000; Becagli et al. 2009; Thomas and Abram, 2016)? Clearly, investigating the mechanisms(s) responsible for MSA migration in polar ice is important if we are to answer these questions and build confidence in the use of [MS⁻] records to infer

past climatic phenomena.

The overarching goal of this study is to develop a better understanding of the environmental and physico-chemical processes that are conducive to MSA migration in cold, polar ice, in order to improve the interpretation of ice-core [MS⁻] records. This paper is organized as follows. In Section 2, we consider published [MS⁻] and ancillary measurements from a variety of Greenlandic and Antarctic ice

cores in an effort to identify any site-specific factors that influence MSA migration. In Section 3, we present observations from the high-resolution, precisely dated DIV2010 ice core (West Antarctica) as a case study for MSA migration. In Section 4, we summarize our current understanding of the physico-chemical processes leading to MSA migration, utilizing an existing model describing soluble impurity transport along ice grain boundaries (Rempel et al., 2002). We derive a simplified version of this model

to illuminate these processes, and to test the ability of different values of MSA diffusivity to reproduce the down-core change in phase relationship between the concentrations of MSA (deposited primarily in summer) and $Na^+$ (deposited primarily in winter) observed along the DIV2010 core. In Section 5, we assess the integrity of the DIV2010 [MS⁻] record, and discuss the broader implications of our results for



the interpretation of [MS⁻] records across a wide range of polar conditions. Section 6 concludes with an overview of our results, and suggestions for future research.

## 2. MSA migration in ice cores

In this section, we use observations to evaluate the relative importance of site-specific variables
on MSA migration in polar ice. We compiled 22 ice core [MS⁻] records originating from 20 sites in Greenland and Antarctica (see Supplementary Material), taken from both the literature and unpublished datasets (Table 1). The following criteria are adopted for selection of the records: (1) high temporal resolution and dating accuracy (annually resolved and dating uncertainty < 3 yr); (2) multi-decadal record length (20 yr minimum); and (3) documented changes in seasonality in the [MS⁻] record and (or)
an explicit mention of MSA migration from summer-to-winter layers. Post-depositional surficial losses of MSA may occur via gaseous diffusion in the top 1-2 meters of the firn at low accumulation sites (Wagnon et al., 1999, Delmas et al., 2003, Weller et al., 2004), As a result, we exclude records from sites where annual mean accumulation rate is less than 100 kg m$^{-2}$ yr$^{-1}$. Also excluded are records from sites subject to moderate to severe melt, which, through percolation, can also rapidly redistribute MS⁻
along the firn (Moore et al., 2005).

Of the 20 sites, thirteen indicate MSA migration, five do not indicate migration, and at two sites migration is deemed unclear at the deepest depths sampled (Fig. 1). The sites and records cover a wide range of climatologic and glaciological conditions, as represented by annual mean surface air temperature, annual mean accumulation rate, distance inland, and impurity concentrations in the ice
(Table 1 and Figure 1). Figure 1a illustrates the range of annual mean accumulation rates ($\dot{b}$) and annual mean surface air temperatures (SAT) of all sites. A strong, nonlinear relationship is apparent between both variables, as expected (Cuffey and Patterson, 2010). Similarly, for the 14 sites where glaciochemical data are available (Table 1), Figures 1b and 1c show, respectively, the depth-averaged concentrations of Na$^+$ ($\overline{Na^+}$; an ionic species whose relevance we expound upon in Sect. 2.2) and MS⁻
($\overline{MS^-}$), taken here as rough measures of winter and summer deposition, respectively, across sites. While impurity deposition largely covaries with distance inland (Fig. 1b and 1c), with higher $\overline{Na^+}$ and $\overline{MS^-}$ at sites nearer to the coast, this pattern does not always hold true, likely due to effects associated with





seasonal atmospheric transport and (or) local differences in the production of the progenitor aerosols of both ions (Iizuka et al., 2016). There is also a notable geographic difference: Greenland values of $\overline{Na^+}$ and $\overline{MS^-}$ (observed at the high-elevation, inland Summit2010 and D4 sites) are lower than those reported for all Antarctic ice cores. In the following sub-sections, we explore in detail the effects of

snow accumulation, ice impurity concentration, surface air temperature, and ice density on MSA migration.

### 2.1 Snow accumulation

Previous work has suggested that high snow accumulation acts as the primary deterrent for MSA migration (Pasteur and Mulvaney, 2000), a result of either (i) the longer distance required for an ion to

travel from the summer to winter layer, (ii) the corresponding suppression of summer-winter layer ionic concentration gradients at higher accumulation sites, or (iii) both (Curran et al., 2002). General support for this suggestion is illustrated in Figure 1a, showing that sites with $\dot{b} > 0.45$ m w. eq. yr$^{-1}$ appear less likely to exhibit clear signs of MSA migration than sites with lower accumulation rates. This consideration, however, does not take into account the depth and (or) time required for migration to

occur. As such, we consider below whether MSA migration invariably occurs across the full range of accumulation rates, given sufficient depth within the firn column.

At low-to-moderate accumulation rate sites ($\dot{b} = 0.1 - 0.45$ m w. eq. yr$^{-1}$), MSA migration seems to universally occur and is observed at increasingly shallow depths at progressively lower accumulation sites (Table 1 and Fig. 2). Data from sites characterized by moderately high accumulation rates ($\dot{b} =$

$0.45 - 0.65$ m w. eq. yr$^{-1}$) are not as straightforward to interpret. For example, in the Dome Summit South (DSS; Law Dome, Antarctica) and Dyer Plateau ice cores, signs of MSA migration were noted in the original studies. At DSS ($\dot{b} = 0.64$ m w.eq. yr$^{-1}$), moderate indications of [MS$^-$] annual maxima extending into adjacent autumn layers were reported at depths lower than about 40-50 m (Curran et al., 2002). Similarly, at Dyer Plateau ($\dot{b} = 0.48$ m w.eq. yr$^{-1}$) [MS$^-$] showed suppressed, localized maxima in

both winter and summer layers throughout a core section from ~51-54 m depth, indicating, perhaps, initial or transitory stages of migration (Pasteur and Mulvaney, 2002). On the other hand, two [MS$^-$] records at sites characterized by annual mean accumulation rates similar to DSS and Dyer, the 2Barrell



($\dot{b} = 0.51$ m w.eq. yr$^{-1}$) and WHG ice cores ($\dot{b} = 0.61$ m w.eq. yr$^{-1}$), respectively, showed no signs of migration. However, neither WHG nor the 2Barrell records extend as deep as their counterparts, suggesting WHG and 2Barrell may be of insufficient length and thus may also show migration at similar depths. Finally, of the four sites characterized by high accumulation rates (>0.65 m w. eq. yr$^{-1}$), three

[MS$^-$] records, the Gomez ($\dot{b} = 0.88$ m. w.eq. yr$^{-1}$), Beethoven ($\dot{b} = 1.2$ m. w.eq. yr$^{-1}$), and DE08 ($\dot{b} = 1.27$ m. w.eq. yr$^{-1}$) ice cores, did not present signs of migration. The exception is the highest accumulation site, Bruce Plateau ($\dot{b} = 1.98$ m. w.eq. yr$^{-1}$), where clear evidence of summer-to-winter migration was reported at ~395 m depth (Porter et al., 2016). Notably, Bruce Plateau is also the deepest drilled record among the high accumulation rate sites of our compilation.

We use these observations to determine the relationship between $\dot{b}$ and the depth of first occurrence of MSA migration, $z_{fo}$, defined as the shallowest reported depth where [MS$^-$] consistently shows its annual maximum in winter (see Supplementary Material for details about the estimation of $z_{fo}$ for each site). We first consider just Antarctic sites reporting MSA migration (Fig. 2a). A least squares fit of a power law of $z_{fo}$ against $\dot{b}$, with the intercept fixed to the origin, yields r$^2$ = 0.99 (Fig.

2a). To ensure that the fit is not dominated by the data pair with the highest $\dot{b}$ value (Bruce Plateau), $z_{fo}$ is regressed against $\dot{b}$ using only those records with accumulation rates < 0.45 m w. eq. yr$^{-1}$, which yields r$^2$ = 0.82 (Fig.2b). Remarkably, extension of the latter fit to the Bruce Plateau data point produces a value of $z_{fo}$ that is within ~15 m of the reported value (Fig. 2a). The ability of the power law to describe the data in both scenarios suggests that, at least at Antarctic sites, the annual mean rate of snow

accumulation has a strong (non-linear) influence on the shallowest depth of MSA migration in the ice column.

Applying this power law to the five sites where MSA migration was not observed or reported (Fig. 1), we find that at three of these sites the core did not reach the predicted $z_{fo}$. The exceptions are (i) the WHG core, where little information on MSA migration in the deepest portions of the core are

provided in the original study (Sinclair et al., 2014), and (ii) the Law Dome DE08 core, where the largest depth reported by Curran et al. (2002) remains within ~15 m of the predicted $z_{fo}$. These results suggest that MSA migration may be more general than commonly thought and that records for which



MSA migration was not reported either may have not reached the requisite depth, or they do not have the necessary resolution, required to observe the phenomenon. The two high elevation records from inland Greenland, which have low overall impurity concentration relative to the coastal Antarctic records, appear anomalous (Fig. 2a). This anomaly may result from the effect of ice impurities – in

particular cationic sea salts – on MSA migration, a point we investigate in more detail below (sections 2.2, 2.5).

## 2.2 Ice Impurities

Earlier studies hypothesized that the presence of well-defined [MS⁻] peaks in winter layers reflects interaction with sea salts, which are preferentially deposited during winter months at most

coastal locations (Legrand and Mayewski, 1997). The attributed mechanism, first formulated by Mulvaney et al. (1992) and echoed by subsequent studies (Wolff et al., 1996; Kreutz et al., 1998, Pasteur and Mulvaney, 2000; Curran et al., 2002), posits that a MSA molecule rejected to and dissolved in the super-cooled liquid veins present at the boundaries of ice grains would be transported by diffusion along a concentration gradient. This mechanism could transport MS⁻ in either the down or up gradient

direction and would not necessarily conform to the concentration gradient measured on bulk ice samples (Sect. 4). Mulvaney et al. (1992) further hypothesized that upon contact with a cation, the precipitation of a stable cation salt through a metathesis reaction could effectively remove MS⁻ from solution, thus sequestering the resultant precipitate in the winter layer.

Here we evaluate whether observations support this mechanism, by comparing $z_{fo}$ to the bulk

concentrations of winter-deposited cations at sites showing evidence of MSA migration. Our evaluation is premised by the notion that higher ionic concentrations have greater ability to regulate the chemical composition of liquid veins, when *in situ* temperatures exceed the eutectic temperatures of the dominant progenitor salt species of those ions (Nye et al., 1991; Section 4). We constrain our analysis to the relationship between MSA and Na⁺ for the following reasons. First, at coastal ice core localities, the

most abundant winter-maximum cation is typically Na⁺, though the relative abundance of $Mg^{2+}$, $Ca^{2+}$, and K⁺ may vary from site to site (Legrand and Mayewski, 1997). Second, past studies have noted that MSA concentration peaks tend to coincide with Na⁺ concentration peaks down-core (Pasteur and



Mulvaney, 2000; Kreutz et al., 1998). Finally, $Na^+$ is considered to be relatively nonreactive within the ice column (Barnes et al., 2003a,b; Legrand and Mayewski, 1997), while less abundant cations, such as $Mg^{2+}$, appear to be more susceptible to post-depositional effects (Kreutz et al., 1998; Wolff et al., 1996).

In Figure 3, $z_{fo}$ is plotted against the core-averaged concentration of $Na^+$ ($\overline{Na^+}$) for all sites where $[Na^+]$ data are available ($n = 11$). These sites have comparable annual mean accumulation rates, within $0.13 - 0.42$ m w. eq. yr$^{-1}$, so that the effect of $\overline{Na^+}$ on MSA migration could be isolated with some confidence. We find that as $\overline{Na^+}$ decreases, MSA migration tends to occur deeper in the firn or ice column (Fig. 3). The depth $z_{fo}$ appears to be particularly sensitive to $\overline{Na^+}$ for small values of $\overline{Na^+}$, as can be seen most prominently for the two Greenland cores showing migration, D4 and Summit2010. This result conforms to the suggestion that, as the concentration of $Na^+$ is reduced, the concentration gradient of MS$^-$ existing in the liquid vein network between the winter and summer layers is also reduced, thereby decreasing the rate at which MSA migration occurs. Another possibility is that the reduction of $[Na^+]$ closes off the liquid vein network within winter layers, inhibiting the vertical transport of ions along grain boundaries, although we lack sufficient data to explore this further.

## 2.3 Temperature

Temperature may influence MSA migration in at least two ways. First, the rate of diffusion of MS$^-$ along grain boundaries may depend on temperature, with higher rates occurring at warmer site, as suggested by Pasteur et al. (1999). Such dependence would imply that sites characterized by lower *in situ* temperatures would tend to exhibit larger values of $z_{fo}$. Second, the precipitation of an MS$^-$-sea salt from solution may be required to sequester MS$^-$ in the winter layer, as proposed by Mulvaney et al. (1992). If effective, the formation of winter-layer $[MS^-]$ annual maxima would be inhibited at sites whose *in situ* temperature exceeds the eutectic temperature(s) of the dominant precipitated MS$^-$-sea salt(s). We can test both of the foregoing thermal influences with the available observations. However, due to a lack of firn temperature profiles, we substitute the more widely available annual mean surface air temperature (SAT) as a surrogate for *in situ* temperature. This approach appears justified, as vertical thermal gradients in polar ice sheets are typically small below 10 m, and their magnitude dictated primarily by local annual mean SAT (Cuffey and Patterson, 2010). We note that deeper in the ice,





thermal gradients could be important (Rempel et al., 2001), particularly near the bedrock due to geothermal fluxes and (or) frictional heat dissipation, but we do not consider those cases here.

As a rough test of a thermal influence on MSA diffusivity, we regress $z_{fo}$ against SAT (not shown). Unlike the relationships between $z_{fo}$ and $\dot{b}$ or $[Na^+]$, no significant relationship between $z_{fo}$ and

SAT is found. Likewise, no significant relationship is found between the $z_{fo}$ residuals of the relationship between $z_{fo}$ and $\dot{b}$ (arguably, the best predictor for $z_{fo}$) and SAT. These results suggest that temperature has a negligible influence on MSA migration, at least for the sites considered (Table 1).

Similarly, available observations do not indicate that sites reporting MSA migration can be discriminated on the basis of SAT. Although sites presenting no evidence for MSA migration tend to

experience high SAT (-19˚C to -12.5˚C, Fig. 1a), two sites where MSA migration has been reported are also characterized by equal or higher SAT. We interpret this observation as an artifact of the strong relationship between SAT and accumulation rate in polar regions (Figure 1), whereupon $\dot{b}$ is deemed to be the more relevant variable in driving $z_{fo}$ (Sect. 2.1). This conclusion, in turn, indicates that either precipitating MS⁻-salts may not drive MSA migration (Mulvaney et al., 1992), or the eutectic

temperature(s) of the dominant MS⁻-sea salt(s) is (are) less than -29.5˚C, the lowest SAT reported in Table 1. We return to this discussion in Section 5.1, after incorporating recent estimates of the eutectic temperatures of the dominant MS⁻-sea salts.

**2.4 Firn or Ice Density**

The mechanism of MSA migration described above, involving diffusion along liquid grain

boundaries, relies on the assumption that the ice grains are sufficiently well compacted to form interconnected premelted veins between the summer and winter layers. This mechanism seems to imply there should be a relationship between the onset of MSA migration and ice density ($\rho$), yet to our knowledge such possible relationship has not been investigated. Here, we consider the firn or ice densities ($\rho_{fo}$) observed at the shallowest depth of MSA migration, $z_{fo}$, again using our compilation of

Greenland and Antarctic data.





In Figure 4, $z_{fo}$ is plotted against $\rho_{fo}$ for the seven sites where available data permit (Table 1). A weighted least squares fit to the data, with the weighting provided by the errors in the $z_{fo}$ estimates, shows a positive relationship between $z_{fo}$ and $\rho_{fo}$ ($p < 0.01$), as expected. Data for low accumulation Berkner Island sites, where annual mean $\dot{b}$ = 0.18 – 0.22 m w.eq. yr$^{-1}$, indicate the occurrence of MSA

migration at densities exceeding about 515-560 kg m$^{-3}$ (Wagenbach et al., 1994), while higher accumulation sites, including THW2010, Ferrigno, and DIV2010 ($\dot{b}$ = 0.28 – 0.41 m w.eq. yr$^{-1}$) show values of $\rho_{fo}$ in the range of 610-650 kg m$^{-3}$. The highest values of $\rho_{fo}$, in the range 710-760 kg m$^{-3}$, are found for the Greenland cores, D4 and Summit2010.

We note that the firn or ice densities at the shallowest depths of MSA migration (~ 500 to ~800

kg m$^{-3}$, Fig. 4), are all comparable to or higher than the critical value of 550 kg m$^{-3}$ that corresponds to the theoretical closest random packing of spherical ice grains (Benson, 1962). Densities greater than 550 kg m$^{-3}$ can be achieved only by bond formation, or sintering, at the contact of individual grains (Cuffey and Patterson, 2010). Persistent vertical transport of super-cooled liquid at the grain boundaries (Mulvaney et al., 1992) requires reaching or exceeding this effective critical density. The data also

indicate that the critical density for MSA migration may vary between sites (Fig. 4), which might be due to variability in grain shape, grain size, and (or) impurity content. For example, firn samples with densities of 350-400 kg m$^{-3}$ can typically be cut into blocks (Cuffey and Patterson, 2010), which suggests that bond formation in the upper firnpack can begin at densities below 550 kg m$^{-3}$. Furthermore, at the lowest accumulation sites, Byrd Station, Siple Dome, and the Filchner-Ronne Ice

Shelf, MSA migration was reported in the shallowest ~5 m of the firn ($z_{fo}$ = 2.6, 2, and 3-4 m, respectively). Although density data are not available for these specific records, this observation suggests that MSA migration may begin at bulk densities substantially lower than 550 kg m$^{-3}$, perhaps through high-density microlayers, or wind-blown crusts, features commonly found at lower accumulation sites (Cuffey and Patterson, 2010). While more data are needed, available evidence

presented here supports the notion that a density of at least 500 kg m$^{-3}$, comparable to the theoretical value of 550 kg m$^{-3}$, is needed for MSA migration.



## 2.5 Synthesis

Our data analysis suggests that the annual mean rate of snow accumulation appears to have a strong influence on the shallowest depth at which MSA migration is observed. The concentration of $Na^+$ in the ice or firn appears to also be an important factor, especially at sites characterized by low

concentrations of $Na^+$. Annual mean SAT appears to play a less important role in determining $z_{fo}$, at least in the SAT range where data are available (-29.5˚C to -12.5˚C). Lastly, the onset of migration appears to be associated with a critical density near 450-550 kg m$^{-3}$, though this may not hold true for all low accumulation sites. Overall, our analysis suggests that MSA migration may be more common than usually thought, and that existing [MS$^-$] records not exhibiting evidence for migration may not be deep

enough, or have the necessary (sub-annual) resolution, to observe the phenomenon.

The shallowest depth of MSA migration ($z_{fo}$) appears most readily predictable from $\dot{b}$ in coastal Antarctica. This result appears in part attributable to the comparably higher concentrations of $Na^+$ in the coastal Antarctic cores than in the two Greenland cores showing evidence of migration (Table 1). The joint effect of annual mean accumulation rate and depth-averaged $Na^+$ concentration on the shallowest

depth of MSA migration is illustrated in Figure 5, which shows MSA migration tends to occur deeper in the firn or ice column for larger values of $\dot{b}$ and lower values of $\overline{Na^+}$. We note that Antarctic sites with comparably low concentrations of $Na^+$ tend to occur further inland and at higher elevations, and thus also tend to have lower accumulation rates than considered in this study (i.e., $\dot{b}$< 0.1 m. w.eq. yr$^{-1}$). Greenland, in contrast, experiences comparably large accumulation rates even at its highest-altitude,

inland locations (e.g., Summit). For a recent overview of Greenland-wide [$Na^+$] deposition, see Rhodes et al. (2017).

## 3. A Case Study: The DIV2010 MSA record

In this section, we use a well-dated multi-century ice core record from DIV2010, an intermediate accumulation rate site in coastal West Antarctica (Table 1), to document in detail the phenomenon of

MSA migration. Prior studies have investigated aspects of the DIV2010 core, including accumulation variability (Medley et al., 2013; Medley et al., 2014) and glaciochemistry (Criscitiello et al., 2013; Criscitiello et al., 2014; Pasteris et al., 2014). As this is the first report of the DIV2010 [MS$^-$] record





below the zone of MSA migration, however, we describe the record in more detail here. Inorganic salt ions (e.g., $Na^+$) and $MS^-$ were measured on discrete samples at a constant sampling interval $\Delta z = 5$ cm using standard suppressed ion chromatography methods (Curran and Palmer, 2001). The age-depth relationship of the core was established independently by identifying summer maxima in three

parameters – [nssS], [$H_2O_2$], and $\delta^{18}O$ – measured at ~2 cm resolution (Pasteris et al., 2014). Here, we examine data from the top 60.4 m of DIV2010 (AD 1905-2010) covering the zone of progressive migration of MSA. Dating uncertainty over this depth interval is estimated to be less than 1 yr based on tie points to well known volcanic events (Pasteris et al., 2014), and the sampling frequency (~10-15 samples per year) allows for seasonally resolved records throughout this depth interval (Criscitiello et

al., 2013; Criscitiello, 2014). At a depth 60.4 m in the core, annual layer thinning estimated using a thinning model assuming frozen basal conditions and a linear vertical strain rate (Nye, 1963) is considerably less than the distance required for a chemical species to migrate from the summer to the winter layer (~30 cm). As a result, no correction for thinning is applied here.

The migration of MSA from summer to winter layers in the DIV2010 ice core appears to be

progressive. To document this progression and contrast the behavior of $Na^+$ and $MS^-$, the month of the annual maximum of [$MS^-$] ($m_{MS^-}$) and the month of the annual maximum of [$Na^+$] ($m_{Na+}$) are each plotted versus age (Fig. 6). The month of the annual maximum of [$MS^-$] tends to change down-core from summer to predominantly winter, as revealed by the significant linear trend of $m_{MS^-}$ between AD 1905-1999 (r = 0.75; p < 0.001). In contrast, [$Na^+$] consistently shows maxima in the winter layers, i.e.,

$m_{Na+}$ portrays no significant trend over this period (r = 0.09, p = 0.41). Similarly, $nssSO_4^{2-}$ displays its annual maximum concentration consistently in the summer layers during this period (not shown). Note that no significant decrease in annual mean [$MS^-$] is found deeper than 10 m, suggesting post-depositional losses are negligible at the DIV2010 core site.

### 3.1 Identification of DIV2010 migration zones

Notwithstanding the progressive nature of MSA migration in the DIV2010 core, we identify three distinct zones in the [$MS^-$] record for this core (Fig.7). First, a *shallow* zone is defined as the depth interval where density is less than 550 kg m$^{-3}$; at DIV2010, this depth occurs at approximately 9.1 m





(Medley et al., 2014). The upper 9.1 m of the core appears to contain the original (unaltered by migration) [MS⁻] variations, to the extent that annual maxima of [MS⁻] are found in the summer layers and are out of phase with winter [Na⁺] maxima. Second, a *transition* zone is defined further down-core, where the [MS⁻] record exhibits no consistent seasonality. Finally, a *deep* zone is defined as the deepest

portion of the record considered, where [MS⁻] and [Na⁺] annual maxima appear to be broadly in phase (Fig.7). In order to facilitate our analysis, we linearly interpolated the [MS⁻] record onto a monthly scale, with the three zones defined to entail an equal number of data points ($n = 132$ months, or 11 years).

To better reveal the variable phase relationship between [MS⁻] and [Na⁺] in the shallow, transition, and deep zones, monthly mean values of [MS⁻] and [Na⁺] are calculated and the standard

error of the monthly means are computed for each zone (Fig.8). In all three zones, Na⁺ shows concentration maxima in winter, albeit with broad variation between April and September. In contrast, MS⁻ exhibits concentration maxima during summer (January) in the shallow zone, but during winter (broadly, May to August) in the deep zone. In the transition zone, the monthly mean [MS⁻] are not significantly different between summer (DJF) and winter (JJA), though local maxima in both spring

(MAM) and fall (SON) are apparent.

### 3.2 Cross-correlation analysis

Cross-correlation coefficients are calculated to quantify the amount of linear relationship between the monthly mean [MS⁻] and [Na⁺] at different lags in the three zones. Due to the finite length of the records, only the cross-correlation coefficients from lag 1 to lag 24 (2 yr) are calculated (Fig. 9).

In the shallow zone, [MS⁻] and [Na⁺] show a negative correlation at lag 0 and a positive correlation at lag 6 (i.e., 6 months); both correlations appear significant at the 5% level compared to two uncorrelated series, of which one is white noise (Chatfield, 1996). These results reflect MSA and Na⁺ being deposited primarily during summer and winter, respectively. In contrast, in the transition zone and more prominently in the deep zone, the [MS⁻] and [Na⁺] records show a positive correlation at lag 0 and a

negative correlation at lag 6; in the deep zone the correlations at lags 0 and 6 are both significant at the 5% level. The positive correlation at zero lag in the transition and deep zones indicates that positive deviations in [MS⁻] tend to coincide with positive deviations in [Na⁺]. The distinct phase relationships





between the two species in the shallow and deep zones are consistent with MSA migration.

## 4. Towards a Mechanistic Understanding of MSA Migration

In this section, we discuss the physico-chemical processes that may be responsible for MSA migration using an existing model of soluble impurity transport. We then derive a linearized version of

the model in order to further illuminate the processes that lead to the movement of MSA from summer to winter layers.

### 4.1 The impurity transport model of Rempel et al. (2002)

The physical mechanisms responsible for the presence of a liquid phase in ice cores, even well below the freezing point of pure water, are well documented (e.g., Nye et al. 1973, Nye 1991) and are

attributed to two distinct physical processes. The first relates to the fact that the atomic radii of most impurity species (not including $F^-$, $Cl^-$, and $NH_4^+$; Wolff, 1996) possess a misfit strain energy that inhibits their incorporation into the tightly packed, crystalline ice lattice. During densification, these impurities are thus preferentially expelled to the boundaries of individual grains of ice. This process has been observed using optical measurements (Mulvaney et al., 1988; Bartels-Rauch et al., 2014). When

the concentration of impurities at the grain boundary is increased, the local equilibrium temperature is decreased, depressing the freezing point of the water-impurity mixture. At temperatures greater than a system's eutectic point, premelted aqueous solutions are assumed to exist at equilibrium as interconnected, submicron veins at the grain boundaries. The second process responsible for the presence of a liquid phase in polar ice pertains to the interstitial curvature occurring at the interface of

three (i.e., triple junctures) or four (i.e., nodes) ice grains. Known as the Gibbs-Thompson effect, this thermodynamic phenomenon is related to the deviation in chemical potential of a vapor surrounding a curved surface from that of the same vapor at equilibrium with a flat liquid surface. In effect, it allows smaller (more curved) ice grains of a given composition to melt at lower temperatures than larger (less curved) ice grains of the same composition (Wettlaufer and Worster, 2006).

Importantly, both of the foregoing processes should respond collaterally to a temperature change, although they may not have the same importance in the maintenance of a liquid phase: scaling





arguments suggest that the curvature effect required to reach a given volume of premelted liquid is negligible in ice sheets in comparison to the effect of impurity-driven super-cooling (Rempel et al., 2001). Focusing on the first process, Rempel et al. (2002) developed an elegant model describing the movement of soluble impurities along crystal grain boundaries. They considered impurity migration due

to (1) temperature gradients (Rempel et al. 2001), which vary gradually downcore (typically <1˚C per 100 m in ice sheet interiors; Cuffey and Patterson, 2010), and (2) impurity concentration gradients, which are characterized by length scales typically on the order of centimeters (Rempel et al. 2002). They derived the following impurity migration equation, referred to below as the RWW model:

$$\frac{\partial}{\partial t} c_{B,k} = -\nabla \cdot (v + v_k)c_{B,k} \ ,$$

(1)

where,

$$v_k \ = \ D_k \frac{\nabla T}{(T_m - T)} + D_k \frac{\sum \Gamma_i \nabla c_{B,i} - \frac{\nabla c_{B,k}}{c_{B,k}} \sum \Gamma_i c_{B,i}}{\sum \Gamma_i c_{B,i}}$$

(2)

Here $c_{B,k}$ is the bulk concentration of the k[th] impurity species, i.e., the mass of the k[th] impurity species per unit ice volume as measured during standard chemical analyses, $t$ is time, and $v_k$ is an effective velocity of the k[th] impurity species relative to the surrounding ice. The ice velocity, $v$, would arise, e.g., from a vertical strain rate and appears to have a small influence on MSA transport on time scales of years to decades (Nye, 1963). As a result, it is systematically neglected in this paper (see also,

Rempel et al., 2002).

As indicated by (2), the relative velocity $v_k$ has two distinct contributions. The first is due to the motion of molecules in the liquid along a temperature gradient and leads to solute transport, even in the absence of concentration gradients. It is proportional to the diffusivity of the k[th] impurity in the liquid, $D_k$, and inversely proportional to the difference $T_m$ -$T$ between the melting point of pure ice ($T_m$) and the

temperature of the liquid (*T*). The second contribution to $v_k$ arises from the concentration gradients of all solutes present in the liquid veins, including the k[th] impurity. As with the first contribution, it is proportional to the diffusivity of the k[th] impurity in the liquid, but it would occur in the absence of a





temperature gradient. It also depends on the slope of the liquidus curve, $\Gamma_i$, of the various solutes that are present in the liquid veins. Scaling arguments suggest that $v_k$ is often well approximated by the second contribution in (2), at least in portions of the ice column where temperature gradients are small (Rempel et al. 2002). The small effect of the first contribution on MSA migration is further supported by

observations made in Section 2.3.

A binary mixture provides the simplest context to discuss the mechanisms of MSA migration in the RWW model. In a mixture comprising $MS^-$ and $Na^+$, equations (1-2) reduce to (for negligible $v$),

$$\frac{\partial}{\partial t} c_{MSA} = -\nabla \cdot (v_{MS} c_{MS}) \ , \tag{3a}$$

$$\frac{\partial}{\partial t} c_{Na} = -\nabla \cdot (v_{Na} c_{Na}) , \tag{3b}$$

where,

$$v_{MSA} = \frac{D_{MS}\Gamma_{Na}}{\Gamma_{MS}c_{MS} + \Gamma_{Na}c_{Na}} \left(\nabla c_{Na} - \frac{c_{Na}}{c_{MS}}\nabla c_{MS}\right) , \tag{4a}$$

$$v_{Na} = \frac{D_{Na}\Gamma_{MS}}{\Gamma_{MS}c_{MS} + \Gamma_{Na}c_{Na}} \left(\nabla c_{MS} - \frac{c_{MS}}{c_{Na}}\nabla c_{Na}\right) . \tag{4b}$$

In this case, the model implicitly assumes that an appreciable amount of the MSA and $Na^+$-containing impurities are rejected from the crystalline lattice of individual ice grains during firnification. The postulated expulsion would concentrate impurities at the grain boundaries, depressing the freezing

point of the inter-granular medium to form sub-micron, super-cooled liquid veins. In general, higher impurity concentrations would lead to higher abundance of premelt liquid. Thus, more premelt liquid is predicted to occur in winter layers, where $[Na^+]$ is typically maximum, than in summer layers, where the comparatively low (bulk) concentration of MSA shows a maximum. The ensuing network of liquid veins would allow the ionic impurities to diffuse under their own concentration gradients, such that a

large proportion of the $MS^-$ from the $MS^-$-rich summer layer migrates to the $MS^-$-poor winter layer. Conversely, a comparatively small proportion of $Na^+$ migrates to the summer layer, because the $Na^+$ concentration difference between the summer and winter layers is reduced by the larger amount of



premelt liquid in the winter layer than in the summer layer. The net result of the different transport rates of MS⁻ and Na⁺ is that variations in [MS⁻] ultimately become in phase with variations of [Na⁺] (Fig.10).

**4.2 Physico-chemical parameters of MSA migration**

The RWW model as applied to the binary system containing MS⁻ and Na⁺ (equations 3-4) includes four parameters: the slopes of the liquidus curve for relevant MS⁻- and Na⁺-containing soluble impurity species ($\Gamma_{MS}$ and $\Gamma_{Na}$), and the grain-boundary diffusivities of MS⁻ and Na⁺ ($D_{MS}$ and $D_{Na}$). Below, we review the existing literature on each of these quantities.

**4.2.1 Grain boundary diffusivity of MS⁻ and Na⁺**

We first consider the grain boundary diffusion coefficient of MS⁻, $D_{MS}$. Rempel et al. (2002), lacking empirical constraints, approximated the diffusivity for ionic constituents in equation (2) as one-third the molecular diffusivity of a bulk liquid (i.e., $D_{MS} = 5 \cdot 10^{-10}$ m² s⁻¹), scaled so as to account for the random orientation of premelted liquid veins in the ice (Lemlich, 1978). Smith et al. (2004) reported a value of $D_{MS} = 2 \cdot 10^{-13}$ m² s⁻¹ for solid ice, estimated by measuring variations in [MS⁻] across horizontal sections of an ice core from Law Dome, Antarctica following nearly 15 years of freezer storage at -20˚C. Using similar ice substrate and experimental set up, Roberts et al. (2009) revised this estimate to $(4.1 \cdot 10^{-13} \pm 2.5 \cdot 10^{-14})$ m² s⁻¹ at -20˚C, which was interpreted by the authors to represent diffusive losses of volatile MSA occurring during extended periods of ice core freezer storage. Notably, this estimate is 1-3 orders of magnitude larger than that reported for solid-state diffusion in ice of HCl, HNO₃, HCHO, and deuteriorated water, despite the molecular radius of MSA greatly exceeding that of each of these species (Roberts et al., 2009). In fact, subsequent studies have contended that the $D_{MS}$ estimate of Roberts et al. (2009) is unlikely to represent pure solid-state diffusion of MSA in firn or ice, and suggested that at least some of the storage-based losses of MSA occurred via liquid transport along grain boundaries (McNeil et al., 2012; Bartels-Rausch et al., 2014). We thus consider the two values of $D_{MS}$ as suggested by Rempel et al. (2002) and Roberts et al. (2009) as potential $D_{MS}$ endmembers.

Although the grain boundary diffusion coefficient for Na⁺, $D_{Na}$, is also under-constrained, empirical evidence supports a relative immobility of Na⁺ in polar ice. For example, Barnes et al. (2003a)





noted no detectable changes in the amplitude of $[Na^+]$ peaks over the past ~11,000 years (top 350 m) in the low-accumulation Dome C ice core record (East Antarctica), while the amplitudes of both $[Cl^-]$ and $[SO_4^{2-}]$ peaks were found to change over the same period. Furthermore, optical measurements at Dome C suggest a predisposition for $Na^+$ to be situated at grain boundaries and for $Cl^-$ to be located

preferentially within the crystalline structure (Barnes et al., 2003b). The importance of these findings is twofold: (i) $Na^+$ appears to be situated in the requisite location to favor the presence of premelt liquid at the grain boundaries, enabling $MS^-$ migration as envisioned in the RWW model to occur; and (ii) $Na^+$ shows greatly reduced mobility relatively to $MS^-$ (or similar sulfur-based acidic species).

### 4.2.2 Liquidus relationships for relevant sea-salt species

The slopes of the liquidus curves, $\Gamma_{MS}$ and $\Gamma_{Na}$, represent linear approximations of the super-cooling as a function of impurity concentration in the liquid phase present near the grain boundaries, $c$, i.e., $T_m - T = \Gamma c$, where $T$ is the *in situ* temperature and $T_m$ is the melting point for pure ice (see Supplementary Material, Fig. S2). Knowledge of $\Gamma$ requires knowledge of the dominant precursor molecular state(s) of the $MS^-$ and $Na^+$ ions present in the ice. Unfortunately, such data remain sparse

beyond those reported in a few notable studies (e.g., Barnes et al., 2003b, Sakurai et al., 2010, Iizuka et al., 2016).

It is generally assumed that all measured $MS^-$ present in polar firn or ice samples derives solely from MSA $(CH_3SO_3H)$ (Sakurai et al., 2010). In the MSA-$H_2O$ system, MSA reaches its eutectic temperature at -75˚C (Stephen and Stephen, 1963). Thus, any MSA molecules expelled to and

concentrated at grain boundaries are expected to exist in liquid solution with $H_2O$. By contrast, $Na^+$ in polar ice may have a number of precursors. For coastal ice cores, however, it seems reasonable to expect that the majority of $Na^+$ is deposited either as NaCl derived primarily from sea spray during storm activity (Legrand and Mayewski, 1997), or as sodium-sulfate salts such as mirabilite, $Na_2SO_4·10H_2O$, derived from brine rejection in sea-ice or from atmospheric sea-salt sulfatization (Rankin et al.,2002;

Iizuka et al., 2016). While the binary system NaCl-$H_2O$ reaches its eutectic at -21.3˚C (Stephen and Stephen, 1963), the eutectic of the $Na_2SO_4$-$H_2O$ system is -1.6˚C (Hougen et al., 1954), suggesting that $Na^+$ deposited as $Na_2SO_4$ should be relatively immobile at most polar ice core sites. Consequentially, the





majority of Na$^+$ relevant to grain boundary migration is likely derived from NaCl. At DIV2010, for example, the molar ratio Cl:Na in the top 60.4 m of the core averages 1.806, similar to the mean molar ratio Cl:Na = 1.798 for seawater (Seinfeld and Pandis, 2006). This indicates a primary marine source of NaCl aerosols at DIV2010, as both brine rejection and sea salt sulfatization in the atmosphere would
tend to produce an offset in the amount of Cl$^-$ deposited (Iizuka et al., 2016).

Although the annual mean ice temperature at most sites showing MSA migration is estimated to be well below -21.3˚C (Sect. 2.3), the upper firn (<10-20 m) undergoes temperature fluctuations that may exceed this value during summer months (Cuffey and Paterson, 2010), thereby providing a potential mechanism to temporarily free Na$^+$ from its bonded state with Cl$^-$. When premelted liquid
solutions containing Na$^+$ and Cl$^-$ refreeze, Cl$^-$ may thus be preferentially allocated within the ice structure (Tokumasu et al., 2015; Barnes et al., 2003b).

With Na$^+$ and MS$^-$ both situated at the grain boundaries, the resulting binary system is the sodium-salt of MS$^-$, CH$_3$SO$_3$Na$\cdot n$H$_2$O, and water (Mulvaney et al., 1992). Recent experimental data indicate that the eutectic temperature for the CH$_3$SO$_3$Na$\cdot n$H$_2$O-H$_2$O system occurs at approximately -
29.3˚C (Sakurai et al., 2010). For comparison, the eutectics for the binary systems Ca(CH$_3$SO$_3$)$_2\cdot n$H$_2$O – H$_2$O and Mg(CH$_3$SO$_3$)$_2\cdot n$H$_2$O – H$_2$O amount to -32.6˚C and -5.0˚C, respectively (Sakurai et al., 2010). Table 2 lists the slopes of the liquidus curves for these MS$^-$-salts in addition to those for alternative relevant sea salts containing Ca$^{2+}$ and Mg$^{2+}$, and SO$_4^{2-}$ (see also Supplementary S2). For the system NaCH$_3$SO$_3$ –H$_2$O, the slope amounts to 6.5 K M$^{-1}$, such that, implicitly, $\Gamma_{MS} = \Gamma_{Na}$.

**4.3 A Simplified Model of MSA Migration**

Although the RWW model provides significant insight into the mechanisms of MSA migration, the system of non-linear partial different equations (1a-b) or (3a-b) does not permit a straightforward analysis (e.g., no closed form solution of these equations with general initial and boundary conditions are available to our knowledge). In this section, we develop a linearized version of the model for the
binary system with MS$^-$ and Na$^+$ (eqs. 3-4) in order to further our understanding of MSA migration. Of course, the insight to be gained is only as reliable as the assumptions upon which the linearized model relies.



Consider the governing equations (3a-b), making it explicit that concentration gradients are strictly vertical,

$$\frac{\partial c_{MS}}{\partial t} = -\frac{\partial}{\partial z}(w_{MS}c_{MS}), \tag{6a}$$

$$\frac{\partial c_{Na}}{\partial t} = -\frac{\partial}{\partial z}(w_{Na}c_{Na}), \tag{6b}$$

where,

$$w_{MS} = D_{MS}\frac{\Gamma_{Na}}{\Gamma_{Na}c_{Na}+\Gamma_{MS}c_{MS}}\left(\frac{\partial c_{Na}}{\partial z} - \frac{c_{Na}}{c_{MS}}\frac{\partial c_{MS}}{\partial z}\right), \tag{7a}$$

$$w_{Na} = D_{Na}\frac{\Gamma_{MS}}{\Gamma_{Na}c_{Na}+\Gamma_{MS}c_{MS}}\left(\frac{\partial c_{MS}}{\partial z} - \frac{c_{MS}}{c_{Na}}\frac{\partial c_{Na}}{\partial z}\right),. \tag{7b}$$

Here, $w_{MS}$ and $w_{Na}$ are the vertical components of MS⁻ and Na⁺ migration velocity, respectively, and $z$ is depth. Three assumptions are made (see also Rempel et al., 2002). First, the slope of the liquidus curve is taken to be the same for the two ionic species, i.e., $\Gamma_{Na} = \Gamma_{MS}$ (section 4.2.2). This

assumption appears plausible if the MS⁻-salt species $CH_3SO_3Na \cdot nH_2O$ dominates in the premelt liquid present near the grain boundaries. With this assumption, the slopes of the liquidus curves cancel out in the defining relationships for $w_{MS}$ and $w_{Na}$ (7a-b). Second, the concentration of MS⁻ is taken to be much smaller than the concentration of Na⁺ in the liquid veins, i.e., $c_{MS} \ll c_{Na}$. This assumption is generally supported by [MS⁻] and [Na⁺] measurements on ice core samples originating from most coastal sites

(section 2.3; Table 1). Under the two assumptions above, relation (7a) becomes

$$w_{MS} = D_{MS}\left(1 + 0\left[\frac{c_{MS}}{c_{Na}}\right]\right)\left(\frac{1}{c_{Na}}\frac{\partial c_{Na}}{\partial z} - \frac{1}{c_{MS}}\frac{\partial c_{MS}}{\partial z}\right), \tag{8}$$

upon expansion of the denominator in a Taylor series. Thus, to the first order in $c_{MS}/c_{Na}$, the speed of

MS⁻ migration can be approximated as





$$w_{MS} = D_{MS} \left( \frac{1}{c_{Na}} \frac{\partial c_{Na}}{\partial z} - \frac{1}{c_{MS}} \frac{\partial c_{MS}}{\partial z} \right). \tag{9a}$$

A similar development for $w_{Na}$ leads to

$$5 \quad w_{Na} = D_{Na} \left( \frac{1}{c_{MS}} \frac{\partial c_{MS}}{\partial z} - \frac{1}{c_{Na}} \frac{\partial c_{Na}}{\partial z} \right) \frac{c_{MS}}{c_{Na}}, \tag{9b}$$

which is also first order in $c_{MS}/c_{Na}$. The ratio of the migration speeds for the two ionic species is thus

$$\frac{w_{Na}}{w_{MS}} = \left( \frac{D_{Na}}{D_{MS}} \right) \frac{c_{MS}}{c_{Na}}. \tag{10}$$

If $D_{MS}$ is comparable to or higher than $D_{Na}$, then Na$^+$ would migrate much more slowly than MS$^-$. In this case, Na$^+$ would be quasi immobile and its concentration at a given depth would vary only slowly with time (compared to MS$^-$). This consideration suggests the following, third assumption. The concentration of Na$^+$ at a given depth in the ice column and at a given time is decomposed into a mean

15  value, $\bar{c}_{Na}(z)$, and a fluctuation, $c'_{Na}(z,t)$,

$$c_{Na}(z,t) = \bar{c}_{Na}(z) + c'_{Na}(z,t). \tag{11}$$

Assuming that $c'_{Na}(z,t) \ll \bar{c}_{Na}(z)$, as suggested by the relatively small mobility of Na$^+$, the vertical

20  speed of MS$^-$ migration along the ice column can be further approximated as

$$w_{MS} = D_{MS} \left( \frac{1}{\bar{c}_{Na}} \frac{\partial \bar{c}_{Na}}{\partial z} - \frac{1}{c_{MS}} \frac{\partial c_{MS}}{\partial z} \right) \tag{12}$$

The insertion of (12) into (6a) yields

$$\frac{\partial c_{MS}}{\partial t} + \frac{\partial}{\partial z}(w_* c_{MS}) = \frac{\partial}{\partial z} \left( D_{MS} \frac{\partial c_{MS}}{\partial z} \right), \tag{13}$$



where $w_*$ is an effective velocity of MS⁻ induced by vertical gradients in [ Na⁺],

$$w_* = \frac{D_{MS}}{\bar{c}_{Na}} \frac{\partial \bar{c}_{Na}}{\partial z} \qquad (14)$$

Thus, under the three stated assumptions, MSA migration can be described by a single, linear partial differential equation (eq. 13) with the MS⁻ diffusivity as a single parameter. In this model (eq. 13), MSA migration arises from two fundamental processes: (1) the convergence or divergence of MS⁻ driven by Na⁺ concentration gradients and (2) the diffusion of MS⁻ along its own concentration gradient. Albeit

10  physically distinct, both processes depend on the diffusivity of MS⁻ in the inter-granular liquid.

It is instructive to consider the character of the steady state distribution of [MS⁻] according to the linearized model. With the tendency term $\partial c_{MS}/\partial t$ set to zero, equation (13) reduces to

$$\frac{\partial}{\partial z}\left(\frac{c_{MS}}{c_{Na}} D_{MS} \frac{\partial \bar{c}_{Na}}{\partial z}\right) = \frac{\partial}{\partial z}\left(D_{MS} \frac{\partial c_{MS}}{\partial z}\right), \qquad (15)$$

given the defining relation for $w_*$ (eq. 14). If $D_{MS}$ is uniform along the ice column ($\partial D_{MS}/\partial z = 0$), and at depths where Na⁺ shows an extremum ($\partial \bar{c}_{Na}/\partial z = 0$), equation (13) becomes

$$\frac{c_{MS}}{\bar{c}_{Na}} \frac{\partial^2 \bar{c}_{Na}}{\partial z^2} = \frac{\partial^2 c_{MS}}{\partial z^2}. \qquad (16)$$

Since concentrations are positive quantities, the concentration ratio on the left-hand side of (16) is always positive, implying that the two second-order derivatives should always have the same sign. Thus, minima (maxima) of MS⁻ concentration will coincide with minima (maxima) of Na⁺ concentration. The [MS⁻] profile, regardless of its initial (i.e., unaltered) character, will evolve so as to become eventually

25  in phase with the [Na⁺] profile. Figure 10 illustrates this evolution of the [MS⁻] profile to steady state as simulated by the linearized model and compares it with the evolution simulated with the RWW model





(see Supplementary S3 for details about the initial conditions, boundary conditions, numerical method of solution, and parameters for these two models).

The following example illuminates the respective roles of the effective velocity, $w_*$, and of the diffusivity, $D_{MS}$, in the MSA migration process. Consider a locally Gaussian profile of $[Na^+]$,

$$\bar{c}_{Na} \propto \exp(-[z - z_0]^2/2\sigma^2) \, , \tag{17}$$

where $z_0$ is the depth where $[Na^+]$ is maximum and $\sigma$ describes the spread of $Na^+$ on each side of the maximum (Fig. 11). In this case, the effective velocity $w_* = -D_{MS-}(z - z_0)/\sigma^2$ is positive above $z_0$

and negative below $z_0$, and the migration equation (13) becomes

$$\frac{\partial c_{MS}}{\partial t} - \frac{D_{MS}}{\sigma^2}(z - z_0)\frac{\partial c_{MS}}{\partial z} = D_{MS}\frac{\partial^2 c_{MS}}{\partial z^2} + \frac{D_{MS}}{\sigma^2}c_{MS} \, , \tag{18}$$

where it has been again assumed that $D_{MS}$ is vertically uniform. Interestingly, the migration equation

(18) has the familiar form of an advection-diffusion-reaction equation. The 2nd term on the left-hand side corresponds to downward advection of MS⁻ above $z_0$ and to upward advection of MS⁻ below $z_0$, i.e., it tends to accumulate MS⁻ at $z_0$. The 1st term on the right-hand side describes Fickian diffusion. Finally, the 2nd term on the right-hand side is a "reaction" term that stems from the vertical variation of $w_*$. It is always positive, effectively leading to MS⁻ production throughout the ice column at a rate proportional

to the amount of MS⁻ initially present. As time progresses, all $[MS^-]$ maxima that may be present in the ice section where $[Na^+]$ is distributed according to (17) will be gradually shifted toward $z = z_0$. At steady state, the $[MS^-]$ profile will be maintained by a balance between MS⁻ advection to the $[MS^-]$ maximum and effective production on the one hand, and the diffusion of MS⁻ away from the $[MS^-]$ maximum on the other hand (as illustrated in Fig. 11).

**4.4 Assessment of MS⁻ Diffusivity**

We next aim to constrain a range of values for MS⁻ diffusivity along ice grain boundaries consistent with observed MS⁻ concentrations in polar ice. To this end, the simplified model of MSA




migration (eq. 13), which includes $D_{MS}$ as the sole parameter, is solved for different values of $D_{MS}$ and model results are compared with data from the DIV2010 ice core introduced above.

The model is solved numerically using finite differences (Supplementary S3; Figures S5 and S6). The model domain, whose upper boundary is situated at $z = 9.1$ m and whose lower boundary is at $z$

$= 60.4$ m, is prescribed to represent the present-day shallow zone at the DIV2010 site. The model grid has a uniform spacing ($\Delta z = 0.05$ m), with grid points coinciding with the sampling depths of the DIV2010 core. The grid cell interfaces coincide with the upper and lower boundaries of the domain. With this configuration of the grid, the boundary conditions of the model consist of a vanishing flux of MS⁻ prescribed at the upper and lower boundaries of the domain:

$$w_* c_{MS} - D_{MS} \frac{\partial c_{MS}}{\partial z} = 0, \qquad \text{at } z = 9.1 \text{ m and } 60.4 \text{ m.} \tag{19}$$

The initial conditions of the model consist of an idealized [MS⁻] profile obtained by linearly interpolating, at the model grid points, the (unaltered) monthly mean [MS⁻] values for the DIV2010

shallow zone (Fig. 5). The vertical profile of [Na⁺], which determines the effective velocity of MS⁻ ($w_*$), is directly derived from the measured profile of [Na⁺] in the shallow zone of DIV2010 core (since model grid points coincide with sampling depths, no interpolation is necessary).

The model is integrated over a time interval that approximates the time it would take for the shallow zone to be buried by a layer of equal thickness through surface accumulation (see

Supplementary S4 for details). This final time, denoted as $t_f$, is taken as ~95 yr, based on the difference between the age of the sample at z = 9.1 m) and the age of the sample at z = 60.4 m. At the end of the model integration ($t = t_f$), the cross-correlation between the [MS⁻] and [Na⁺] profiles simulated by the model between z = 9.1 and 60.4 m is calculated and compared to the cross-correlation between the measured [Na⁺] and [MS⁻] profiles over the same depth interval. This procedure is repeated for four

different values of $D_{MS}$, $10^{-10}$, $10^{-11}$, $10^{-12}$, and $10^{-13}$ m² s⁻¹, encompassing the values assumed or suggested in prior studies (e.g., Rempel et al., 2002; Roberts et al., 2009). A "good" value of $D_{MS}$ is expected to lead to a "good" agreement between the simulated and observed cross-correlations, at least at small lags.





Consider first the model solution with $D_{MS} = 10^{-13}$ m$^2$ s$^{-1}$. At $t = t_f$, the simulated [MS$^-$] profile has not significantly deviated from its initial profile, which is negatively correlated at zero lag with the [Na$^+$] profile (Fig.12a). This result suggests that the value of $D_{MS} = 10^{-13}$ m$^2$ s$^{-1}$ is too small to account for the down-core change in phase relationship between [Na$^+$] and [MS$^-$] observed at DIV2010. For $10^{-12}$ m$^2$ s$^{-1} < D_{MS} < 10^{-11}$ m$^2$ s$^{-1}$, the simulated cross-correlation at zero lag between [Na$^+$] and [MS$^-$] switches from negative to positive. For values of $D_{MS} \geq 10^{-11}$ m$^2$ s$^{-1}$, it is positive but much stronger than observed, suggesting that these values may be too large. Thus, the $D_{MS}$ value that best explains DIV2010 data would be in the range from $10^{-12}$ to $10^{-11}$ m$^2$ s$^{-1}$, i.e., greater than the value of $(4.1 \cdot 10^{-13} \pm 2.5 \cdot 10^{-14})$ m$^2$s$^{-1}$ reported by Roberts et al., (2009) and lower than the value of $5 \cdot 10^{-10}$ m$^2$ s$^{-1}$ assumed by Rempel et al. (2002). We stress that this result is immune to potential dating errors in the sense that the cross-correlation coefficients are calculated for different vertical spacings along the core, not for different time lags; calculating cross-correlations at different time lags leads to a similar result (Fig. 12b; see also Supplementary Figures S5 and S6).

While the $D_{MS}$ range estimated by best fitting the DIV2010 data is instructive, we note it is not necessarily universal, as grain boundary diffusivities are expected to vary in response to multiple glaciological factors. For example, the experimental results of Kim et al. (2008) show that the diffusion coefficients of ions in super-cooled mixtures are a function of both ionic concentration and temperature. Additionally, physical properties of the firn and ice, including porosity, density, and grain size, may affect the partitioning of chemical impurities to grain boundaries (Spaulding et al., 2011). Even at a given site, seasonal and interannual variations in impurity concentrations may lead to down-core changes in $D_{MS}$. Nonetheless, the approach presented in this section suggests that a range of $D_{MS}$ values can be directly derived from [MS$^-$] measurements along a given ice core.

## 5 Paleoclimatic Implications

### 5.1 Revisiting the effect of temperature on grain boundary migration

In section 2.3, we tested prior hypotheses stating that post-depositional formation of winter [MS$^-$] maxima occurs as a result of the precipitation of MS$^-$-salts from their grain boundary solutions in sea-salt rich winter layers (Mulvaney et al., 1992; Curran et al., 2002). While such a mechanism suggests




MSA migration could occur only at sites where *in situ* temperatures do not exceed the eutectic temperature(s) of the relevant MS⁻-salt(s), our data compilation did not support such a distinction (Fig. 1; section 2.3). Here we revisit this conclusion, equipped with the mechanistic framework provided by the RWW model.

At 19 of the 20 sites considered in in our study (Table 1), the estimated ice temperature exceeds the eutectic value (-29.3°C) for the sodium salt of MS⁻, $CH_3SO_3NaH_2O$. At these 19 sites, MSA migration would be expected to occur following the model of Rempel et al. (2002), whereby migrating species are implicitly assumed to remain dissolved in the grain boundary liquid. The only site where ice temperature is estimated to be less than -29.3°C is the Summit2010 record from Greenland (Maseli et

al., 2017), where MSA migration is also observed but not predicted to occur based on this model. We offer two explanations for this sole discrepancy. First, the reported annual mean SAT (Table 1, Giese et al., 2015) falls just below, but very close to, the eutectic temperature for this system (within 0.2°C), suggesting *in situ* mean annual ice temperatures exceeding -29.3°C below 10 m depth are possible. An alternative possibility is that the $Ca^{2+}$-salt of MS⁻ (which has a lower eutectic of -32.6°C) may be more

important than the sodium salt of MS⁻ in driving MSA migration at Summit2010, given the much higher abundances of $Ca^{2+}$ in inland Greenland ice compared to coastal Antarctica (Iizuka et al., 2008).

     A eutectic effect is further supported by the observation that no discernable MSA migration occurs in the subannually-resolved portion (i.e., down to ~10.5 m) of the [MS⁻] record from South Pole (SP-95), a site with annual mean SAT of -51°C (Meyerson et al., 2002). While SP-95 was not

considered in our data compilation due to the site's low $\dot{b}$ (0.08 m w.eq. yr⁻¹), the lack of clear MSA migration at SP-95 departs from the expected relationship found between $\dot{b}$ and $z_{fo}$ in Antarctica (Sect. 2.1; Fig. 2). This suggests that MS⁻ at SP-95 may be immobilized through a metathesis reaction once partitioned to the grain boundaries, a reaction which need occur only in winter layers.

     In sum, we propose that sites will only be subject to potentially-severe MSA migration if *in situ*

temperatures exceed the relevant eutectic temperatures, which will allow MS⁻ to remain as a solution even in the presence of $Na^+$ and $Ca^{2+}$ cations situated at the grain boundaries. Such temperatures are inferred here to range from -29.3°C to -32.6°C, depending on the binary system considered (Sakurai et al., 2010). Importantly, most coastal ice core locations, despite their paleoclimatic significance, are



susceptible to MSA migration given their relatively high SAT (Table 1), although high accumulation rates will mitigate this phenomenon to some degree (section 2.1). One coastal region of exception may be northeastern Greenland, where relatively cold SATs (approximately -30 to 33°C; Weißbach et al., 2016) may keep the ice below the eutectic temperature of the primary $MS^-$-salts even in near-coastal

environments.

**5.2 Vertical extent of MSA migration**

A reigning question in the use of $[MS^-]$ in polar ice as a paleoclimate proxy is the extent of MSA migration along the core. Past studies have circumvented this potential issue by assuming either that (1) MSA migration is confined within an annual layer (Kreutz et al., 1998; Curran et al., 2003; Thomas and

Abram, 2016), or (2) multi-year averages of $[MS^-]$ are largely unaffected by migration (Wolff et al., 1996). While (2) appears a more conservative approach, the requisite averaging period, and thus the maximum resolution that can be achieved in a paleoclimatic reconstruction given MSA migration remain unknown. In this section, we examine assumptions (1-2) using DIV2010 data and the linearized model of MSA migration.

In an effort to account for a range of initial (i.e., unperturbed) $[MS^-]$ profiles in the shallow zone of the DIV2010 ice core (Sect. 3.1), a large number (10,000) of numerical experiments of MSA migration are conducted. The initial $[MS^-]$ profile of a given experiment is obtained by adding, to the monthly mean $[MS^-]$ values observed in the shallow zone of DIV2010, a normal noise with a mean of zero and a variance equal to that of the shallow zone mean monthly values. If a negative concentration

value arises in the initial profile, the procedure is repeated until all values in the profile are positive. Using this approach, inter-annual variability in the initial $[MS^-]$ profile is emulated, such that no two years should contain the same mean $[MS^-]$ in a given experiment, nor should a given year display the same mean $[MS^-]$ for different experiments. In contrast, all experiments rely on the same $[Na^+]$ profile measured for DIV2010. Given the uncertainties in $MS^-$ diffusivity (section 4.4), two sets of experiments

are considered: a first set with $D_{MS} = 10^{-12}$ m$^2$ s$^{-1}$ and a second with $D_{MS} = 10^{-11}$ m$^2$ s$^{-1}$ (so that 2 x 10,000 = 20,000 experiments are actually performed). For all experiments, the model is subjected to a condition of no $MS^-$ flux at both the upper and lower boundaries of the domain, and is integrated over a





time interval $t = t_f = 95$ yr (section 4.4).

Figure 13a shows (i) the [Na$^+$] profile observed at DIV2010 and used to constrain the effective velocities $w_*$ in the model, and (ii) the annual mean [MS$^-$] profiles simulated by the model at $t = t_f$. For $D_{MS} = 10^{-11}$ m$^2$ s$^{-1}$, the changes in annual mean [MS$^-$] relative to the initial [MS$^-$] profile are much larger

than for $D_{MS} = 10^{-12}$ m$^2$ s$^{-1}$ (Fig. 13b), particularly prior to AD 1975 ($t = 25$ yrs). In some sections of the simulated profiles, dramatic positive or negative changes in annual mean [MS$^-$] occur, depending on the magnitude of the local [Na$^+$] gradients. In some individual years (e.g., AD 1954), relative changes in the annual mean [MS$^-$] reach 60 to 100%, clearly negating the first assumption above, that MSA migration is confined within an annual layer.

Given this finding, we next explore whether and over what range a multi-year average of [MS$^-$] data could be used to more accurately reflect the original (i.e., unaltered) multi-year mean [MS$^-$] concentration. To this end, we average the simulated [MS$^-$] profiles at $t = t_f$ in different time intervals ranging from 3 to 15 years and compare the final (altered) averages to the initial (unaltered) averages (Fig. 13c). As expected, the difference between the final and initial averages of [MS$^-$] decreases as the

averaging period increases. The difference shows only modest reduction as the averaging period increases from 7 to 15 years.

In sum, while our results may pertain only to DIV2010 and rely on a series of modelling assumptions (section 4.3), two points appear worthy of note. First, for [MS$^-$] records showing evidence of MSA migration, the assumption that MSA remained confined within annual layers may not be

generally valid, given in particular the high inter-annual variability in the concentrations of Na$^+$ and other major impurities potentially conducive to MSA migration, which is typical of most ice cores originating from coastal sites (Legrand and Mayewski, 1997). High inter-annual variability in [Na$^+$], for example, corresponds to large vertical [Na$^+$] gradients along the core, which tend to increase the anomalous diffusion of MS$^-$ within the super-cooled liquid present at grain boundaries. Second, at least

for [MS$^-$] records exhibiting severe MSA migration such as at DIV2010, averaging the data over a time period of approximately ten years may constitute a reasonable compromise between accuracy and temporal resolution for paleoclimatic reconstruction.



### 5.3 Revisiting the combined influence of snow accumulation and [Na$^+$] on MSA migration

In Section 2, we provided empirical evidence that two local factors appear to influence the shallowest depth of MSA migration in polar ice cores: annual mean accumulation rate and core-averaged Na$^+$ concentration. Here, we use a model of soluble impurity transport to assess whether the

ability to predict $z_{fo}$ from these two factors is also mechanistically grounded. Since the linearized model is not valid for small values for [Na$^+$]/[MS$^-$], the original model of Rempel et al. (2002) (Section 4.1) is used in an effort to produce results of more general validity.

We first simulate, for a range of accumulation rates and depth-averaged [Na$^+$] values, the time it takes for a [MS$^-$] maximum present in an annual layer and initially out of phase with the [Na$^+$]

maximum in the layer ($\varphi = 180^o$), to align with the [Na$^+$] maximum in the layer ($\varphi = 0^o$). Given the asymptotic nature of the concentration evolutions simulated by the model, we approximate this time as the time at which the phase difference between the [MS$^-$] and [Na$^+$] maxima drops to $\varphi < 20^o$. Experiments are conducted for different extents of the model domain, to represent different values of annual layer thicknesses. For each experiment, the initial [MS$^-$] and [Na$^+$] profiles in the layer are

sinusoidal functions of depth, with (i) a period set equal to the layer thickness, (ii) a [MS$^-$] maximum present in the middle of the layer, and (iii) two [Na$^+$] maxima present at the top and bottom of the layer (see Supplementary S5 for details). The model is subjected to a condition of no flux both at the top and at the bottom of the layer. The model parameters are set to $\Gamma_{MS} = \Gamma_{Na} = 6.5$ K mol$^{-1}$ and $D_{MS-} = D_{Na+} = 10^{-11}$ m$^2$ s$^{-1}$ or $10^{-12}$ m$^2$ s$^{-1}$.

The time required for approximate phase alignment ($\varphi < 20^o$) of the simulated [MS$^-$] and [Na$^+$] profiles in the layer are shown for two different layer thicknesses and a range of layer-averaged [Na$^+$] values (Fig. 14a); this time is referred to as $t_\varphi$ below. Two notable conclusions can be drawn. First, it is seen that $t_\varphi$ increases with layer thickness and decreases with layer-averaged [Na$^+$]. Similar results are displayed in Figure 14b in a form that is reminiscent of Figure 5 (scaled linearly) and Figure 14c (scaled

logarithmically), which both show the combined effect of annual mean accumulation and core-averaged [Na$^+$] on the shallowest depths of MSA migration in our data compilation. To the extent that annual layer thickness increases with annual mean accumulation rate, the model results appear to be





qualitatively consistent with the data and provide a theoretical basis to the notion that $\dot{b}$ and $[Na^+]$ could be used to predict $z_{f0}$. Both current observations and the present set of experiments with the RWW model suggest that $[MS^-]$ records from ice cores characterized by high accumulation and low core-averaged $[Na^+]$ should undergo relatively small alteration by MSA migration.

## 5 6. Conclusions

Polar ice core records of methanesulfonic acid have been used to draw inferences about oceanic and atmospheric processes at polar latitudes on a range of time scales. However, both observation and theory suggest that MSA is mobile in the ice column, leading to uncertainties about its integrity as an indicator of past climatic conditions. Here, we synthesize existing data from a range of polar

environments and consider an impurity transport model to study MSA migration in polar ice. Emphasis is placed on (i) the environmental conditions that favor MSA migration and (ii) a better understanding of the physico-chemical processes causing the movement of MSA in polar firn and ice.

Our analysis shows that the shallowest depth at which MSA migration occurs in coastal ice cores varies primarily with annual mean accumulation rate. In Antarctica in particular, a power law

characterizes this relationship accurately. It suggests that the absence of MSA migration observed in some ice cores from high accumulation sites stems from the fact that chemical measurements for these cores have not reached depths sufficient to have undergone migration. This observation leads us to conclude that MSA migration in polar ice is more general than commonly thought. Annual mean surface air temperature and the concentration of the dominant cation sea salt, $Na^+$, appear to be less influential

than accumulation rate under most circumstances, at least at most coastal Antarctic sites and in the temperature range from -29.5˚C to -12.5˚C. A notable exception is for inland Greenland sites, where MSA migration tends to occur deeper in the core than would be predicted by the power law, an offset hypothesized to stem from extremely low concentrations of marine-derived impurities relative to most coastal Antarctic sites. Our analysis further suggests that MSA migration generally takes place once firn

or ice density reaches a critical value near 550 kg m$^{-3}$, which corresponds to the tightest packing of spherical ice grains in the firn and enabling the formation of premelted liquid veins at grain boundaries. However, at some low accumulation sites ($\dot{b} = 0.1 - 0.2$ m w. eq. yr$^{-1}$), MSA migration is seen to occur



at depths where bulk density is likely to be less than 550 kg m$^{-3}$. This result suggests that small-scale variability in ice density is important and (or) that other factors may also determine the onset of MSA along the firn or ice column.

New high-resolution data from the West Antarctic DIV2010 ice core shows annual [MS$^{-}$] maxima gradually shifting down-core from austral summer, when MSA deposition is high, to austral winter, when MSA deposition is low and Na$^{+}$ deposition is high. As a result, a down-core change in the phase relationship between [MS$^{-}$] and [Na$^{+}$] is observed, whereby [MS$^{-}$] and [Na$^{+}$] are negatively correlated at zero lag in the upper part of the core and positively correlated at zero lag in the lower part of the core, providing evidence of the progressive nature of MSA migration.

A linearized version of the impurity transport model of Rempel et al. (2002) is derived for a binary mixture containing MS$^{-}$ and Na$^{+}$ in order to further understanding of the MSA migration phenomenon in polar ice. In this linearized model, MS$^{-}$ transport is governed by a single linear partial differential equation with MS$^{-}$ diffusivity in super-cooled liquid ($D_{MS}$) as the sole parameter. In this model, MSA migration arises from two transport processes: (1) the convergence or divergence of MS$^{-}$

driven by [Na$^{+}$] gradients and (2) the diffusion of MS$^{-}$ along its own concentration gradient. Analysis of this model shows that [MS$^{-}$] maxima (minima) are bound to coincide with [Na$^{+}$] maxima (minima) along the ice column, regardless of the timing of MSA deposition maxima. The model, therefore, provides a mechanistic explanation for the tendency for MSA, deposited mainly during summer, to present concentration peaks in winter layers in the deepest part of polar ice cores.

Finally, we use the linearized MSA migration model and the DIV2010 data to gain insight into two poorly constrained yet critically important aspects of MSA migration. First, we evaluate different values of the grain boundary diffusivity of MS$^{-}$, $D_{MS}$. We find that $D_{MS}$ values in the range from $10^{-12}$ to $10^{-11}$ m$^{2}$ s$^{-1}$ lead to the most accurate simulations of the down-core change in the phase relationship between [MS$^{-}$] and [Na$^{+}$] observed at DIV2010. Second, using this range of values, we apply the model

to investigate the extent to which MSA migration has altered the original [MS$^{-}$] record for DIV2010. We estimate the errors incurred by averaging [MS$^{-}$] data over annual (and multi-year) intervals, an approach often adopted to reduce the effect of migration on the interpretation of [MS$^{-}$] records. We find that MSA migration can lead to significant changes in the annual and multi-year [MS$^{-}$] averages. This result





suggests that [MS⁻] records severely perturbed by MSA migration may best be used to infer decadal and lower-frequency climate variability, though further investigation using a range of [MS⁻] records and a better constrained model is needed to investigate this further.

The migration of MSA in cold, polar ice is a fascinating but challenging phenomenon. This paper covers many, but not all, of its observational and theoretical aspects. For example, contentions of MSA migrating away from regions of high acidity in the core, as caused by the deposition of compounds of volcanic origin (Curran et al., 2002; Delmas et al., 2003), are not explored here. While the model of Rempel et al. (2002) provides an important mechanistic framework for understanding MSA migration in polar ice and perhaps for ultimately correcting its effects for paleoclimatic reconstruction, its usefulness remains limited by uncertainties about key physico-chemical parameters. These include most notably the diffusivities of the relevant migrating species in super-cooled liquid, the slope of the liquidus curves for relevant, interacting species, and the partitioning of impurities between the ice lattice and the surface of the ice grains. Laboratory studies under a range of controlled conditions would help constrain these parameters, improve our understanding of MSA migration in polar ice, and make full use of the paleoclimatic potential of this compound.

**Data Availability**

The DIV2010 chemistry data used in the generation of this manuscript is available on request from M.O.

**Author Contribution**

M. Osman, S. B. Das, and O. Marchal designed the study. M. Osman carried out the data synthesis. O. Marchal developed the linearized model and M. Osman coded the RWW model and the linearized model. S. B. Das collected the DIV2010 ice core. M. J. Evans measured the DIV2010 ice core ion data. M. Osman, S. B. Das and O. Marchal interpreted the results. M. Osman wrote the paper. All co-authors commented on the paper.

**Competing interests**



The authors declare that they have no competing conflicts of interest.

**Acknowledgements:** We thank Elizabeth Thomas for contributing the Ferrigno data, Joseph McConnell for contributing the Summit2010 and D4 data, and Alison Criscitiello and Weifu Guo for early

discussions about MSA migration in the DIV2010 record. M. B. Osman acknowledges government support awarded by DoD, Air Force Office of Scientific Research, National Defense Science and Engineering Graduate (NDSEG) Fellowship, 32 CFR 168a. This work was supported by the US NSF (ANT-0632031 and PLR-1205196 to S. B. Das, and NSF-MRI-1126217 to M. J. Evans. and a Woods Hole Oceanographic Institution Interdisciplinary Research award to S. B. Das. and O. Marchal.

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

**Figures**





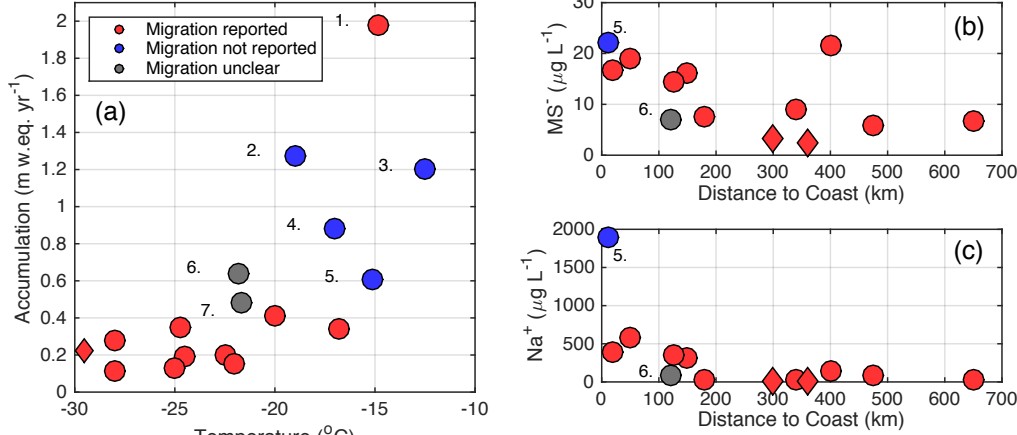

**Figure 1: Compilation of data for Antarctic (circles) and Greenland (diamonds) ice cores for which [MS⁻] records meet the criteria of this study. (a) Annual mean accumulation rate versus annual mean surface air temperature. (b) Core-averaged [MS⁻] versus distance to the coast. (c) Core-averaged [Na⁺] versus distance to the coast. The colors indicate whether MSA migration has been reported, deemed as unclear, or not reported in the original publications. The numbers indicate ice core sites with $\dot{b} > 0.45$ m w. eq. yr⁻¹: 1. Bruce Plateau (length of record: 448 m), 2. DS08-Law Dome (196 m, corresponding to a time span of 145 yr), 3. Beethoven Plateau (47 m, 28 yr), 4. Gomez Nunutak (56 m, 42 yr), 5. WHG-Victoria Land (106 m, 130 yr), 6. DSS-Law Dome (124 m, 156 yr), 7. Dyer Plateau (80 m, 103 yr).**

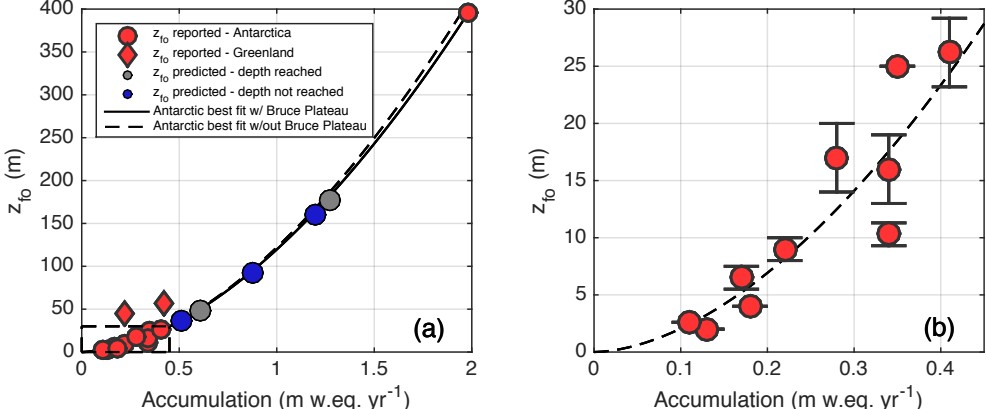

**Figure 2: (a) Shallowest depth of MSA migration, $z_{fo}$, versus annual mean accumulation rate ($\dot{b}$) for Antarctic (circles) and Greenland sites (diamonds). The dashed line is the least-squares fit $z_{fo} = 125 \cdot \dot{b}^{1.77}$ through the Antarctic data with $\dot{b} < 0.45$ m. w .eq. yr⁻¹ (n = 10; r² = 0.82) and used to predict $z_{fo}$ at sites where MSA migration was not reported (grey and blue). At 2 of these sites (WHG and DE08; grey), maximum sampling depth exceeds the predicted $z_{fo}$. The solid black line is the fit when including the Bruce Plateau data ($\dot{b}$ = 1.98 m. w .eq. yr⁻¹). (b) Expanded view of panel (a) (rectangle bounded with dashed lines). Error estimates**




for $z_{fo}$ are generally crude (Supplementary S1). Some are based on unit conversion from m w. eq. to m using a firn densification model (Harron and Langway, 1980) constrained by site diagnostic observations (Table 1) and an assumed surface snow density range of 300-400 kg m$^{-3}$. The dashed line is the same as in panel (a).

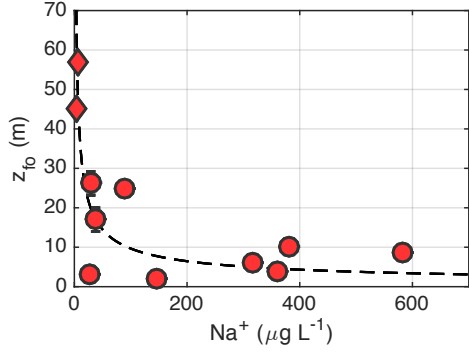

Figure 3. Shallowest depth of MSA migration versus core-averaged [Na$^+$] for different core sites in Antarctica (circles) and Greenland (diamonds). The dashed line is the least-squares fit $z_{fo}$ = 142·[Na$^+$]$^{-0.58}$ through all the data (n = 11, r$^2$ = 0.85).

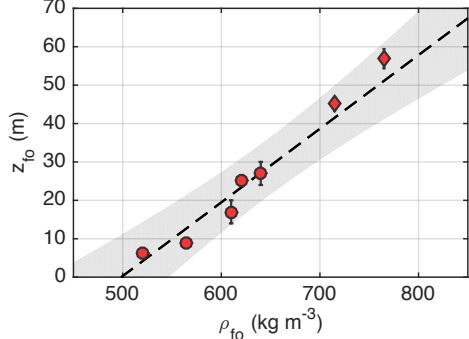

Figure 4. Shallowest depth of MSA migration versus firn or ice density for different core sites in Antarctica (circles) and
15   Greenland (diamonds). The vertical bars are error estimates (Supplementary S1), the dashed line is the weighted least-squares fit $z_{fo} = -95.6 + 0.192\rho$ through the data (n = 7; r$^2$ = 0.89), with the weighting provided by the $z_{fo}$ error estimates, and the shaded region is the region of 95% confidence.





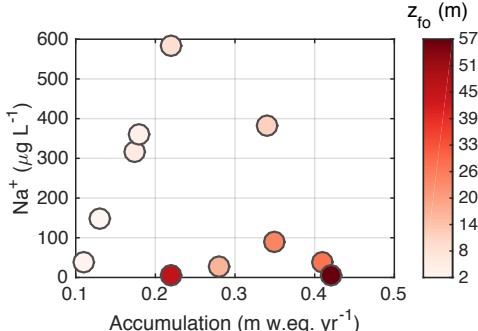

**Figure 5. Core-averaged [Na$^+$] versus annual mean accumulation rate for different core sites in Antarctica and Greenland. The different colors correspond to different values of the shallowest depth of MSA migration.**

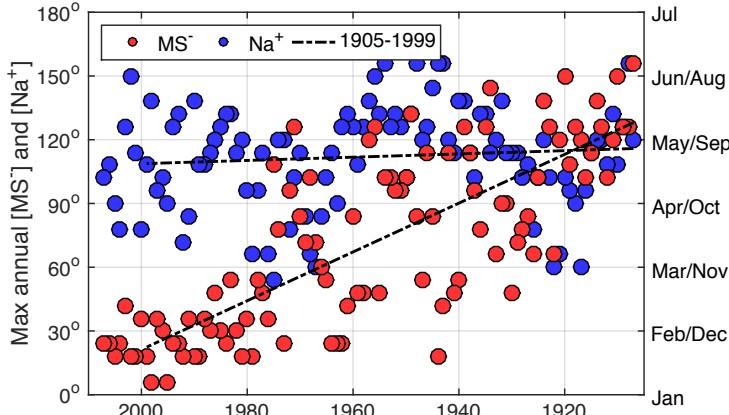

**Figure 6: Timing of annual maximum of [MS$^-$] (red) and annual maximum of [Na$^+$] (blue) versus calendar age (year AD) for the DIV2010 core. No obvious predisposition for migration in the up or down-core direction is observed in DIV2010. As such, the**
10 **timing of annual maximum [MS$^-$] and annual maximum [Na$^+$] are defined in terms of the number of degrees out of phase of January 1$^{st}$ (0°), where 180° indicates an annual maximum on July 1$^{st}$. The two dashed lines are the least-squares fits for [MS$^-$] and [Na$^+$] over the period AD 1999-1905, corresponding to depths below which $\rho = 550$ kg m$^{-3}$. The fit is highly significant for [MS$^-$] (n = 95, r = 0.75, p < 0.0001) and not significant for [Na$^+$] (n = 95, r = 0.09, p = 0.41).**





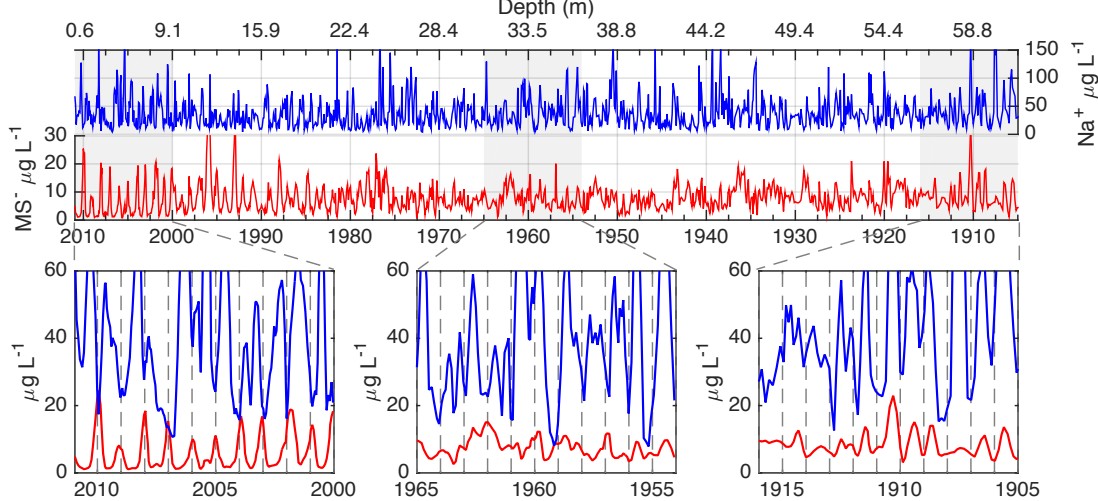

**Figure 7: Records of [MS⁻] (red) and [Na⁺] (blue) from the DIV2010 ice core. Upper panel: the entire record considered in this study, with a depth scale at the top and a time scale (year AD) at the bottom (raw data). Bottom panels: 11-yr long portions of the records within the shallow zone (left), the transition zone (middle), and the deep zone (right). Three-point running averages are displayed for each zone. Dashed vertical lines denote January 1ˢᵗ of each year.**

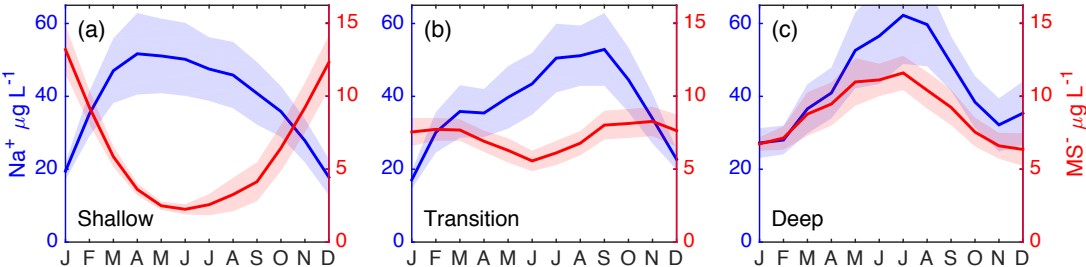

**Figure 8: Monthly mean concentration of Na⁺ and MS⁻ in the shallow (a), transition (b), and deep (c) zones of DIV2010. The shaded regions indicate ± 1 standard error of the mean.**



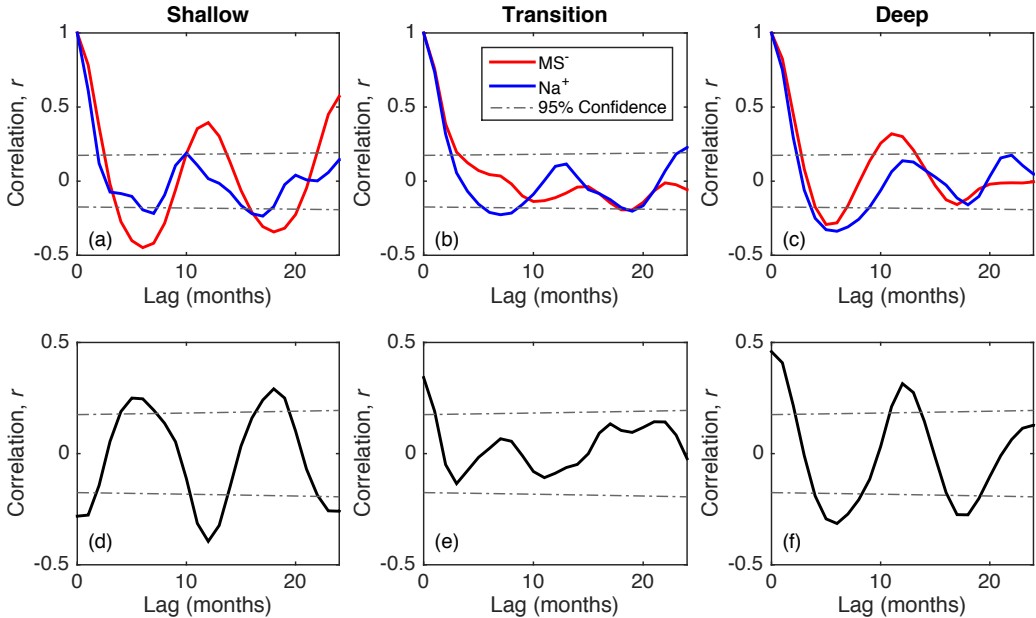

**Figure 9:** (a-c) Correlograms of [MS⁻] and [Na⁺] in the three zones of the DIV2010 ice core. The horizontal dashed lines show the 95% confidence interval for a time series with no autocorrelation (white sequence). (d-f) Cross-correlation between [MS⁻] and [Na⁺] in the three zones of the DIV2010 ice core. The horizontal dashed lines show the 95% confidence interval for two uncorrelated time series.

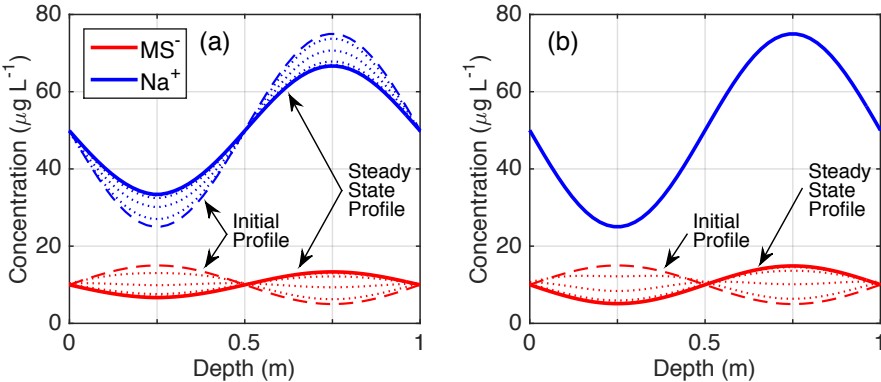

**Figure 10:** Comparison between the RWW model and the linearized model. (a) Profiles of [MS⁻] and [Na⁺] simulated by the RWW model with $D_{MS} = D_{Na} = 10^{-11}$ m² s⁻¹ and $\Gamma_{MS} = \Gamma_{MS} = 6.5$ K M⁻¹. (b) Profiles of [MS⁻] simulated by the linearized model with an effective velocity based on the [Na⁺] profile shown in the panel and with $D_{MS} = 10^{-11}$ m² s⁻¹ (see text and Supplementary for details).



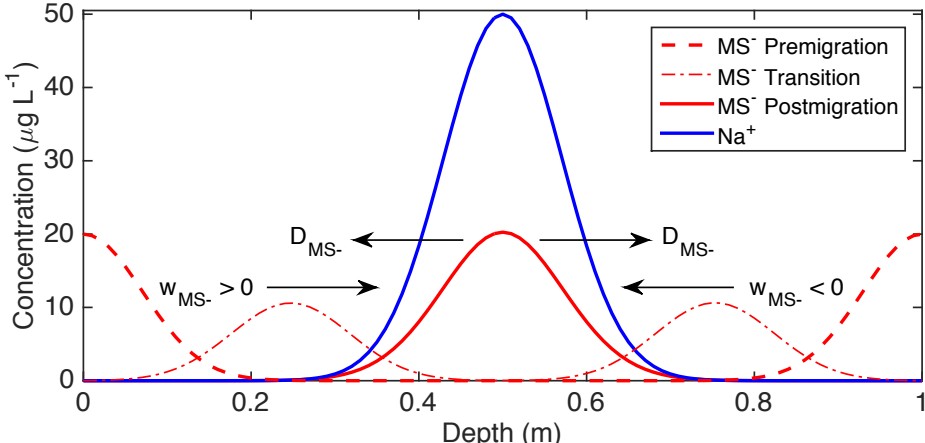

**Figure 11: Evolution of [MS⁻] simulated by the linearized model assuming (i) an effective velocity based on a Gaussian profile of [Na⁺] (blue) and (ii) $D_{MS} = 10^{-11}$ m² s⁻¹. The dashed, dotted-dashed, and solid red lines show, respectively, the initial, transient, and steady state profiles of [MS⁻]. The different symbols and arrow show the two transport processes affecting the steady-state profile of [MS⁻]: MS⁻ convergence toward the [Na⁺] maximum, and MS⁻ diffusion away from the [Na⁺] maximum (see text and Supplementary Material for details).**

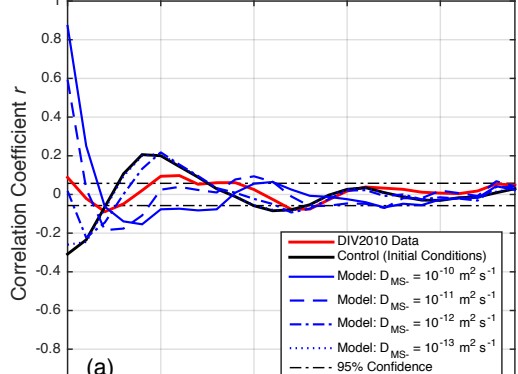
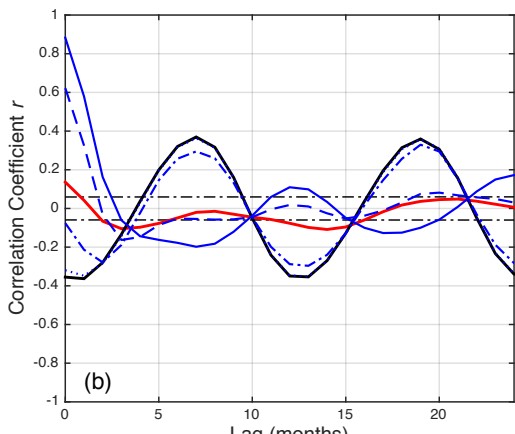

**Figure 12: Cross-correlation between [Na⁺] and [MS⁻] at different vertical spacings (a) and different time lags (b). In each panel, the red line shows the cross-correlations calculated from the DIV2010 data below the shallow zone (z > 9.1 m), the blue lines show the cross-correlations calculated from the linearized model for different values of $D_{MS}$, the solid black line shows the cross-correlations corresponding to the initial conditions of the model, and the horizontal dashed lines show the 95% confidence interval for two uncorrelated time series (see text and Supplementary for details).**



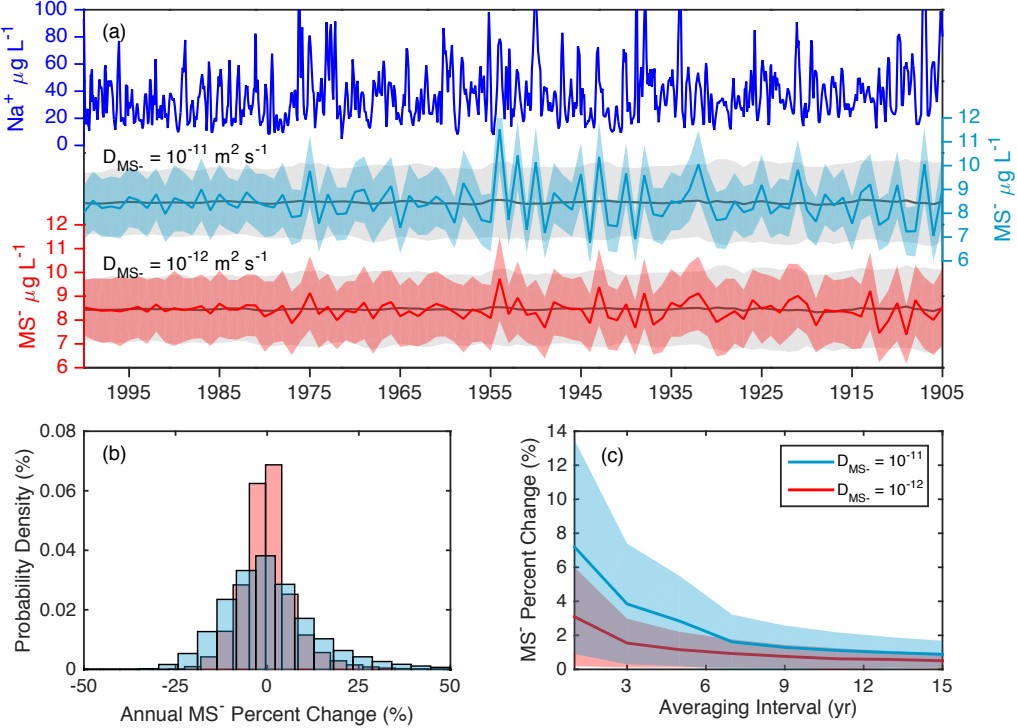

**Figure 13: Extent of MSA migration in the DIV2010 ice core as estimated from the linearized model. Panel (a), top: DIV2010 [Na⁺]**
**record (5-point smoothed). Panel (a), middle: (i) Annual means of [MS⁻] (black line) and standard deviations of [MS⁻], $\sigma$ (gray**
5 **band, representing $\pm 1\sigma$ about the means) for the initial conditions (ICs) of the model experiments, and (ii) annual means of [MS-]**
**(blue line) and standard deviations of [MS⁻] (blue band) for the terminal conditions (TCs, ~ 95 yr) of the model experiments for**
$D_{MS} = 10^{-11}$ m² s⁻¹. **Panel (a), bottom: Same as panel (a), middle, but for** $D_{MS} = 10^{-12}$ m² s⁻¹. **Panel (b): Probability density of the**
**absolute difference between the annual mean [MS⁻] for the ICs and TCs, normalized to the annual mean [MS⁻] of the ICs, for all**
**model experiments with** $D_{MS} = 10^{-11}$ m² s⁻¹ **(blue) and** $10^{-12}$ m² s⁻¹ **(red). Panel (c): Absolute difference between the annual mean**
10 **[MS⁻] for the ICs and TCs, normalized to the annual mean [MS⁻] of the ICs, for all model experiments with** $D_{MS} = 10^{-11}$ m² s⁻¹ **(blue)**
**and** $10^{-12}$ m² s⁻¹ **(red), as a function of the data averaging interval. The shaded regions illustrate the dispersion ($\pm 1$ standard**
**deviation) of the normalized absolute differences among the model experiments.**



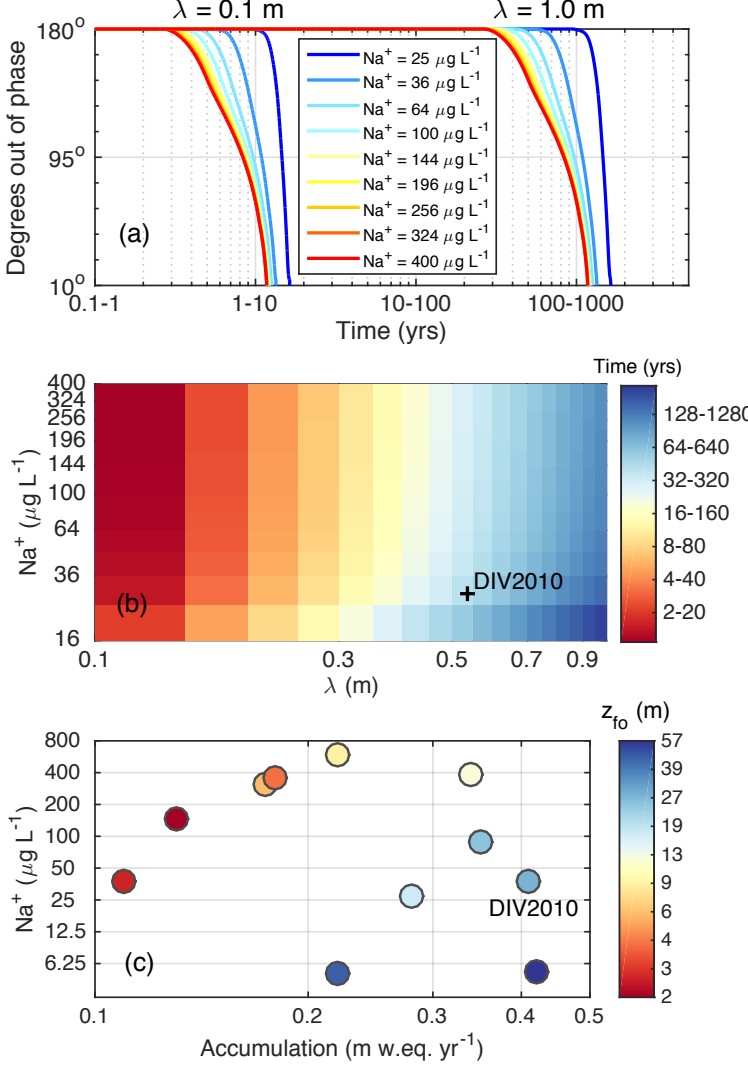

Figure 14: (a) Evolution of the phase difference between [MS⁻] and [Na⁺] maxima for (i) two different extents of the model domain or "annual layer thickness" (0.1 and 1 m) and (ii) different layer averages of [Na⁺] (25 – 400 µg L⁻¹), as calculated from the RWW model. Note the two different scales along the horizontal axis: the scale from 0.1 to 100 yr applies to model results with $D_{MS}$ = 10⁻¹¹ m² s⁻¹, and the scale from 1 to 1,000 yr applies to model results with $D_{MS⁻}$ = 10⁻¹² m² s⁻¹. (b) Time required for approximate alignment of [MS⁻] and [Na⁺] maxima for different values of "annual layer thickness" and different layer averages of [Na⁺], as calculated from the RWW model. Conditions for the DIV2010 core site are indicated by the cross. (c) Shallowest depth of MSA migration for different values of annual mean accumulation rate and different core averages of [Na⁺] according to our data compilation, with DIV2010 denoted. Note the logarithmic scales in all panels.



## Tables

**Table 1. Ice core sites where MSA migration (or lack thereof) has been reported ("~" denotes unavailable data and "n/a" denotes a non-applicable field).**

| Ice core name | Lat/Lon | Year collected | Depth reached/ reported, m (yrs) | Elevation, m | Distance to coast, km | SAT, ˚C | $\dot{b}$, m w. eq. $yr^{-1}$ | MSA migration reported?* | $z_{fo}$, m (yr)*** | $\rho_{fo}$, kg $m^{-3}$ | $[\overline{MS^-}]$, µg $L^{-1}$ | $[\overline{Na^+}]$, µg $L^{-1}$ | Reference |
|---|---|---|---|---|---|---|---|---|---|---|---|---|---|
| West Antarctic MSA records | | | | | | | | | | | | | |
| Dyer Plateau | 70.65˚S, 65.02˚W | 1988 – 1989 | 56 m w.eq. (103) | 1943 | 190 | -21.7 | 0.48 | U | n/a | ~ | ~ | ~ | Mulvaney et al., 1992; Pasteur and Mulvaney, 2000 |
| Dolleman Island (1) | 70.58˚S, 60.93˚W | 1985 – 1986 | 96 m w.eq. (297) | 398 | 20 | -16.8 | 0.34 | Y | 10 (10) | ~ | 16.6 | 382 | Mulvaney et al., 1992 |
| Dolleman Island (2) | 70.58˚S, 60.93˚W | 1992 – 1993 | 18.5 m w.eq. (45) | 398 | 20 | -16.8 | 0.34 | Y | 8.5 m w.eq. (19) | ~ | ~ | ~ | Pasteur and Mulvaney, 2000 |
| Gomez Nunatak | 74.02˚S, 70.63˚W | 1980 – 1981 | 37 m w.eq. (42) | 1130 | 135 | -17 | 0.88 | N | n/a | n/a | ~ | ~ | Mulvaney and Peel 1998; Pasteur and Mulvaney, 2000 |
| Beethoven Peninsula | 71.88˚S, 74.57˚W | 1992 – 1993 | 30 m w.eq. (28) | 580 | 16 | -12.5 | 1.20 | N | n/a | n/a | ~ | ~ | Pasteur and Mulvaney, 2000 |
| Berkner Island North (1) | 78.30˚S, 46.28˚W | 1989 – 1990 | 11 (21) | 730 | 50 | -22.5 | 0.22 | Y | 9 (16) | 560 | 19.0 | 583*** * | Wagenbach et al., 1994 |
| Berkner Island North (2) | 78.30˚S, 46.28˚W | 1994 – 1995 | 39 m w.eq. (174) | 730 | 50 | -22.5 | 0.20 | Y | ~ | ~ | ~ | ~ | Pasteur and Mulvaney, 2000 |
| Berkner Island South | 79.60˚S, 45.62˚W | 1989 – 1990 | 11 (28) | 940 | 150 | -24.5 | 0.17 | Y | 6 (16) | 520 | 16.0 | 317*** * | Wagenbach et al., 1994 |
| Siple 94-1 | 81.65˚S, 148.8˚W | 1994 – 1995 | 148 (1150) | 621 | 400 | -25 | 0.13 | Y | 2 (6) | ~ | 21.7 | 147 | Kreutz et al., 1998 |
| DIV2010 | 76.80˚S, 101.7˚W | 2010 – 2011 | 112 (216) | 1329 | 180 | -24 | 0.41 | Y | 26 (38) | 640 | 7.7 | 37.6 | Criscitiello et al., 2014; Criscitiello, 2014 |
| THW2010 | 77.0˚S, 121.2˚W | 2010 – 2011 | 62 (145) | 2020 | 340 | -28 | 0.28 | Y | 17 (32) | 610 | 9.0 | 27.5 | Criscitiello et al., 2014; Criscitiello, 2014 |
| Bruce Plateau | 66.03˚S, 64.07˚W | 2009 – 2010 | 448 (~) | 1976 | 30 | -14.8 | 1.98** | Y | 395 (560) | ~ | ~ | ~ | Goodwin, 2013; Porter et al., 2016 |
| Byrd NBY-2 | 80.02˚S, 119.5˚W | 1989 – 1990 | 164 (629) | 1530 | 650 | -28 | 0.11 | Y | 2.6 (13) | ~ | 6.7 | 37 | Langway et al., 1994 |
| Filchner-Ronne D235 | 77˚S, 64˚W | 1987 – 1988 | 4.3 m w.eq. (20) | ~ | 125 | ~ | 0.18 | Y | 1.8 m w.eq. (9) | ~ | 14.4 | 361*** | Minikin et al., 1994 |
| Ferrigno | 74.57˚S, 86.90˚W | 2010 – 2011 | 136 (309) | 1354 | 475 | -24.7 | 0.35 | Y | 25 (30) | 620 | 5.9 | 88 | Thomas et al., 2016 |
| East Antarctic MSA records | | | | | | | | | | | | | |
| Law | 66.77˚S, | 1997 | 45 (191) | 1370 | 110 | -22 | 0.15 | Y | ~ | ~ | ~ | ~ | Curran et |





| | | | | | | | | | | | | | |
|---|---|---|---|---|---|---|---|---|---|---|---|---|---|
| Dome: W20k | 112.35°E | | | | | | | | | | | | al., 2002 |
| Law Dome: DSS | 66.77°S, 112.42°E | 1997; 2000 | 124 (156) | 1370 | 120 | -21.8 | 0.64 | U | n/a | n/a | 7.0 | 85.5 | Curran et al., 2002; Curran et al., 2003 |
| Law Dome: DE08 | 66.72°S, 113.18°E | 1986 | 196 (145) | ~ | 100 | -19 | 1.27 | N | n/a | n/a | ~ | ~ | Curran et al., 2002 |
| WHG - Victoria Land | 72.90°S, 169.08°E | 2006 | 105 (130) | 400 | 12 | -15 | 0.61 | N | n/a | n/a | 22 | 1901 | Sinclair et al., 2012; Sinclair et al., 2014 |
| **Greenland MSA records** | | | | | | | | | | | | | |
| Summit2010 | 72.33°N, 38.28°W | 2010 | 87 (268) | 3213 | 360 | -29.5 | 0.22 | Y | 43.7-46.9 (120-130) | 710-720 | 3.4 | 5.1***** | Maselli et al., 2017 |
| D4 | 71.4°N, 44.0°W | 2004 | 145 (270) | 2710 | 300 | ~ | 0.42 | Y | 54.3-59.4 (90-100) | 760-770 | 2.5 | 5.3**** | MSA data unpublished |
| 2Barrell | 76.94°N, 63.15°W | 2011 | 21.3 (21) | 1685 | 100 | ~ | 0.51 | N | n/a | n/a | ~ | ~ | Osterberg et al., 2015 |

\* Y = Yes; N = No; U = Unclear

\*\* Bruce Plateau annual mean accumulation rate appears to be highly variable over the core-depth; listed value represents the AD 1750-2010 estimate (Goodwin, 2013).

\*\*\* See Supplementary S1 and Table S1 for more information on how $z_{fo}$ is defined at each site.

\*\*\*\* $[\overline{Na^+}]$ estimated using the reported site value for $[\overline{Cl^-}]$, and converted assuming a site-ratio $[\overline{Cl^-}]/[\overline{Na^+}] = 1.798$, identical to mean sea-water ratio (Seinfeld and Pandis, 2006).

\*\*\*\*\* $[\overline{Na}]$ measurements made using continuous-flow inductively coupled plasma mass spectrometry (ICP-MS), as opposed to $[\overline{Na^+}]$ measurements made using ion chromatography (IC). As ICP-MS measures both the soluble and insoluble mass content, $[\overline{Na}]$ values are likely slightly higher than IC-based estimates of $[\overline{Na^+}]$.

**Table 2. Slopes of the liquidus curves (Γ) estimated for various binary mixtures composed of impurity species of likely relevance and water. All values listed in columns prior to the Γ column are required for calculation of Γ (see Supplementary S2).**

| Species | Mol. mass (g mol⁻¹) | Eutectic temperature (°C; binary with H₂O) | Eutectic composition (wt%; binary with H₂O) | Density at eutectic (g mL⁻¹; *=approximated) | Γ (K M⁻¹) | Reference |
|---|---|---|---|---|---|---|
| $CH_3SO_3H$ | 96.11 | -75.0 | 51.1 | 1.20 | 11.7 | Stephen and Stephen, 1963 |
| $Na(CH_3SO_3)$ | 119.11 | -29.3 | 47 | 1.15* | 6.5 | Sakurai et al., 2010 |
| $Mg(CH_3SO_3)_2$ | 214.50 | -5.0 | 14.2 | 1.15* | 6.6 | Sakurai et al., 2010 |
| $Ca(CH_3SO_3)_2$ | 230.27 | -32.6 | 47 | 1.20* | 12.9 | Sakurai et al., 2010 |
| $NaCl$ | 58.44 | -21.3 | 23.3 | 1.16 | 4.6 | Stephen and Stephen, 1963 |
| $MgCl_2$ | 95.21 | -33.0 | 21.6 | 1.13 | 12.9 | Stephen and |




| | | | | | | |
|---|---|---|---|---|---|---|
| | | | | | | Stephen, 1963 |
| $CaCl_2$ | 110.98 | -51.0 | 30 | 1.19 | 15.9 | Stephen and Stephen, 1963 |
| $H_2SO_4$ | 98.08 | -62.0 | 35.6 | 1.19 | 14.3 | Hornung et al., 1956 |
| $Na_2SO_4$ | 142.04 | -1.6 | 4.0 | 1.12 | 5.5 | Hougen et al., 1954 |
| $MgSO_4$ | 120.37 | -3.6 | 17.3 | 1.22 | 2.3 | Marion et al., 1999 |
| $CaSO_4$ | 136.14 | -0.7 | 18.0 | 1.23 | 0.4 | Rolnick, 1954 |