# Peer review of "Methanesulfonic acid (MSA) migration in polar ice: Data synthesis and theory"

_The Cryosphere, 2017_

## Referee Comment (RC1) · A. Rempel (Referee) · 10 Jun 2017

General Comments

The concentrations of trace constituents measured in polar ice cores record changes in conditions at the time of deposition at the glacier surface. However, there is clear empirical evidence (e.g. from anomalies of volcanic origin that exhibit increasingly gradual onsets with age, changing seasonality of MSA peaks) that some degree of post-depositional redistribution can take place. The current manuscript examines the movement of MSA signals in considerable detail with a combination of empirical data and theoretical analysis applied primarily to a new high resolution dataset from the DIV2010 core. The results of this effort include important new constraints on the envi-

ronmental variables that are most important for determining the depth at which significant MSA migration can take place, an informative linearized model that predicts the evolution of MSA concentration in response to the changes in liquid content imparted by seasonal variations in the impurity loading that are gauged by Na concentrations, and a new determination of the effective diffusivity of MSA that is held responsible for the concentration changes observed in the DIV2010 core. This represents substantial progress beyond previous understanding of impurity migration in polar ice, and will help in the interpretation and design of future sampling efforts.

Specific Comments

The manuscript is well organized and clearly written. I appreciated the examination of site-specific variables contributing to MSA migration, including regression analyses leading to best-fit relationships (figs. 2-4) with the depth at which migration is evident. If the authors could provide some further intuition for the source of the exponents in these power laws, this would be a useful addition to the synthesis subsection (2.5). The description of the DIV2010 MSA record and related variables in section 3 is succinct and informative. The mechanistic treatment of MSA migration is particularly clear and represents an important advance over earlier work, particularly by providing constraints on the effective diffusivity of MSA in the DIV2010 core. The value obtained for this key parameter is one or two orders of magnitude smaller than that typically used to describe compositional diffusion in pure water; this might suggest a significant role for motion along two-grain boundaries rather than only in the liquid veins that line triple-junctions and their associated nodes at 4-grain intersections. The authors appear to have made a conscious decision not to speculate on the details of the precise migration pathways, referring only to "grain-boundary" migration rather than specifying whether they expect the vein–node network or the two-grain boundaries to dominate. A brief comment on the distinctions between these possibilities might be of use for some readers. The paleoclimatic implications are well summarized in the final substantial section of the

paper, prior to the conclusions.

Technical Corrections

I didn't notice many typos or other technical issues requiring the authors' attention. The term "super-cooling" is used throughout, whereas previous authors have taken care to use "under-cooling" instead since super-cooling is most commonly used to refer to liquid in a transient, disequilibrium state. Line 5 of page 27 repeats the word 'in' twice.
* * *

---

## Referee Comment (RC2) · Anonymous Referee #2 · 27 Jul 2017

The manuscript by Osman et al describes a very detailed study on the migration of MSA in ice cores. A crucial finding is correlation between the depth at which migration is observed and the accumulation site, asking for caution when concluding the absence of MSA migration in short cores. Further, the role of a number of physical parameters and processes on MSA migration is discussed in detail, it turns out that accumulation rate is an important one. Last but not least, diffusivity coefficients are suggested based on detailed modelling work on a high resolution ice core.

I'm impressed how the study combines expertise in ice core analysis with fundamental physical chemistry. Therefore, it is of paramount interest to a wide scientific audience. The manuscript is very long due to to the wealth of information and the carful and precise description of the analysis. It reads very charming, the conclusions are well

justified, and assumptions and uncertainties in the analysis are openly mentioned. I refrain from recommending immediate publication, because I need clarification regarding the fundamental aspect in discussing and applying the phase diagrams. These fundamental details are directly linked to the conclusion of the manuscript and one of the questions raised in the introduction "Why should MSA in particular exhibit migrations, while associated soluble impurities and acids do not?".

My concern comes down to the point, that I can't follow how the impurity transport model by Rempel as presented in the manuscript leads to a transport from summer to winter layers via concentration driven diffusion. I can think of two scenarios:

(1) At T above -20°C, isolated patches of NaCl (winter) and NaCl/MSA (summer) solution form. If temperature increases, volume of the liquid brine increases. At a specific T the two patches might meet. If they do not mix, MSA will diffuse from summer to winter resulting in a constant concentration. On first approximation NaCl might have the same concentration in both patches, so it does not diffuse. This scenario will not build up a new peak at the winter location in the core, but rather smooth the MSA over the whole year.

(2) At T between -75 and -30°C (MSA is still in solution, but NaCl and NaMS are solid). Thus a liquid patch at summer location holding only MSA will form. If that spreads or moves it might meet NaCl crystals. There, crystallisation of NaMS could occur, which will built up a concertation gradient and lead to diffusion of more and more MSA towards the winter layer. This might indeed lead to a complete shift of the MSA peak from summer to winter. But, is the diffusion of MSA in liquid rate determining, or the spreading/movement of the film, or the solution of NaCl, or the precipitation of NaMS?

I attached a graph for illustration. It is very likely that I miss an important point here, but may I ask you to clarify the ultimate process in more detail? As a third option, could MSA also be pushed into the gas phase from solution and be transported by gas-phase diffusion? May I ask you to comment on this aspect. (M. H. Kuo, S. G. Moussa and V.

F. McNeill, Atmos. Chem. Phys., 2011, 11, 9971–9982.)

Second, I would like to read more about the grain boundary net-work along which migration of the MSA takes place. A) What is the crystal size, grain boundary density at the position where MSA migration is observed? B) The diffusivity that is discussed is then an effective diffusivity in a porous medium like water/sand or air/snow. When comparing diffusivity between different ice cores or between single crystals and ice cores the grain boundary density (or its volume fraction) needs to be taken into account. I acknowledge that –taken the missing data- this is not possible, but would encourage a more detailed discussion on this issue (F. Dominé, M. R. Albert, T. Huthwelker, H.-W. Jacobi, A. A. Kokhanovsky, M. Lehning, G. Picard and W. R. Simpson, Atmos. Chem. Phys., 2008, 8, 171–208.)

Minor comments

Page 7 line 10 ff: Here I wonder, if the observation of MSA migration at these depth is a matter of time rather than ice density at that depth. Time is mention in the intro to this 2.1 but then I miss a discussion or final conclusion on time.

Page 18 line 4 ff: "The RWW model as applied to the binary system containing MS- and NA+". This confuses me. The binary system is water-NaMS. Or, do we have a ternary system water-NaCl-MSA? Connected to this: Page 19 line 10: How can an ion have a liquidus curve? The phase diagram is different for each counter ion.

Page 18 line 18: I suggest to add and discuss the work of Domine on diffusion in single crystals (E. Thibert, E. Thibert and F. Dominé, J. Phys. Chem. B, 1998, 102, 4432–4439; F. Dominé and E. Thibert, J. Phys. Chem. B, 1997, 101, 3554–3565.) and discuss the role of grain boundaries with respect to diffusion in porous media in more detail (last point hold througout the text).

I hope you find these comments helpful and I'm looking forward to your revised manuscript.

Please also note the supplement to this comment:
https://www.the-cryosphere-discuss.net/tc-2017-84/tc-2017-84-RC2-supplement.pdf

**Supplement:**

A) $T > |-20°C$  (NaCl ; Na MS ; HMS Liquid)

Na⁺ Cl⁻   Na⁺ Cl⁻
H⁺ MS⁻

ICE

SUMMER        WINTER

$T \nearrow$

NaCl⁻   Na⁺ Cl⁻
H⁺MS⁻

MS⁻ diffuse
H⁺

Na⁺ H⁺   Cl⁻ MS⁻

B) $T > -75°C < -30°C$   (NaCl solid ; NaMS solid ; MSA liquid)

H⁺MS⁻        ◇ ◇

ice

$T \nearrow$

H⁺      MS⁻
◇ ◇

MSA ↓

H⁺   Cl⁻
○ ○        NaMS

---

## Author Comment (AC1) · 8 Sep 2017

A. Rempel (Referee)

rempel@uoregon.edu

*Dear Dr. Rempel,*

*We thank you for your valuable feedback. We have reviewed all your comments/suggestions, and have attempted to sequentially address each to the best of our ability. For convenience, your comments are reproduced below, our replies to them are in italics, and excerpts from the text are between double quotes.*

**General Comments**

The concentrations of trace constituents measured in polar ice cores record changes in conditions at the time of deposition at the glacier surface. However, there is clear empirical evidence (e.g. from anomalies of volcanic origin that exhibit increasingly gradual onsets with age, changing seasonality of MSA peaks) that some degree of post-depositional redistribution can take place. The current manuscript examines the movement of MSA signals in considerable detail with a combination of empirical data and theoretical analysis applied primarily to a new high-resolution dataset from the DIV2010 core. The results of this effort include important new constraints on the environmental variables that are most important for determining the depth at which significant MSA migration can take place, an informative linearized model that predicts the evolution of MSA concentration in response to the changes in liquid content imparted by seasonal variations in the impurity loading that are gauged by Na concentrations, and a new determination of the effective diffusivity of MSA that is held responsible for the concentration changes observed in the DIV2010 core. This represents substantial progress beyond previous understanding of impurity migration in polar ice, and will help in the interpretation and design of future sampling efforts.

**Specific Comments**

C1) The manuscript is well organized and clearly written. I appreciated the examination of site-specific variables contributing to MSA migration, including regression analyses leading to best-fit relationships (figs. 2-4) with the depth at which migration is evident. If the authors could provide some further intuition for the source of the exponents in these power laws, this would be a useful addition to the synthesis subsection (2.5).

*The power law $z_{fo} \propto \dot{b}^E$ with E = 1.77 > 1 (Fig. 2) implies that the depth of first occurrence of MSA migration, $z_{fo}$, is more sensitive to annual mean accumulation rate, $\dot{b}$, for small $\dot{b}$ than for high $\dot{b}$. Interestingly, results from the model of Rempel et al. (2002) (section 5.3) leads to a power law between the time scale of MSA migration, $t_\varphi$, and the layer thickness over which the migration takes place, $\lambda$ (see our new Figure 15, added to the main text and also appended below). That is, $t_\varphi \propto = a\lambda^e$, where $e \approx 2$. Thus, to the extent that $t_\varphi$ and $\lambda$ can be taken as surrogates for, respectively, $z_{fo}$ and $\dot{b}$, the model results appear to be consistent with the data. The observed variation of the shallowest depth of MSA migration with accumulation rate would thus reflect the mere fat that, similarly to Fickian diffusion with constant diffusivity, the time scale for anomalous diffusion ($t_\varphi$) varies about quadratically with the thickness over which the diffusion takes place ($\lambda$). These points are now elaborated in the manuscript in Section 5.3 (Pages 31-32, Lines 27 and 1-5):*

*"The nonlinear relationship found between $z_{f0}$ and $\dot{b}$ (Section 2) is also worth further exploration. Results from the RWW model can be well-approximated by the power law $t_\varphi \propto \lambda^2$ (Fig. 15), which is reminiscent of the power law relationship $z_{fo} \propto \lambda^{1.77}$ derived from our data compilation (Fig. 2). The observed variation of the shallowest depth of MSA migration with accumulation rate would thus reflect the mere fact that, similarly to Fickian diffusion with constant diffusivity, the time scale for anomalous diffusion ($t_\varphi$) varies quadratically with the thickness over which the diffusion takes place ($\lambda$)."*

*(Page 52, Lines 1-6)*

[Figure]

*"**Figure 15: The time required for approximate alignment of [MS⁻] and [Na⁺] maxima ($t_\varphi$) as a function of annual layer thickness ($\lambda$) for different layer averages of [Na⁺] and $D_{MS} = 10^{-11}$ m² s⁻¹. The various curves are the least-squares power law fit $t_\varphi = a\lambda^e$. The exponent e is estimated to about 2 for all values of layer-averaged [Na⁺] (note that the least squares power law fit for $D_{MS} = 10^{-12}$ m² s⁻¹, not shown, yields a value of a that is a factor of 10 higher than for $D_{MS} = 10^{-11}$ m² s⁻¹).**"*

C2) The description of the DIV2010 MSA record and related variables in section 3 is succinct and informative. The mechanistic treatment of MSA migration is particularly clear and represents an important advance over earlier work, particularly by providing constraints on the effective diffusivity of MSA in the DIV2010 core. The value obtained for this key parameter is one or two orders of magnitude smaller than that typically used to describe compositional diffusion in pure water; this might suggest a significant role for motion along two-grain boundaries rather than only in the liquid veins that line triple-junctions and their associated nodes at 4-grain intersections. The authors appear to have made a conscious decision not to speculate on the details of the precise migration pathways, referring only to "grain-boundary" migration rather than specifying whether they expect the vein–node network or the two-grain boundaries to dominate. A brief comment on the distinctions between these possibilities might be of use for some readers.

*We have added a clarifying statement along these lines to Sect. 4.4 (Pages 26-27, lines 28-29 and 1):*

*"...(our derived) $D_{MS}$ does not take into account whether MS⁻ migration is dominated by diffusion at two-grain boundaries, or at the triple junctures and node networks (Wettlaufer and Worster, 2006, Riche et al., 2012)."*

The paleoclimatic implications are well summarized in the final substantial section of the paper,

prior to the conclusions.

**Technical Corrections**

C3) I didn't notice many typos or other technical issues requiring the authors' attention. The term "super-cooling" is used throughout, whereas previous authors have taken care to use "under-cooling" instead since super-cooling is most commonly used to refer to liquid in a transient, disequilibrium state.

*All prior occurrences of "super-cooling" have now been changed to "under-cooling".*

C4) Line 5 of page 27 repeats the word 'in' twice.

*The typo has been corrected (pg. 27, line 19).*

*References*

*Riche, F., Bartels-Rausch, T., Schreiber, S., Ammann, M., and Schneebeli, M.: Temporal evolution of surface and grain boundary area in artificial ice beads and implications for snow chemistry, J. Glaciol., 58, 815–817, doi:10.3189/2012JoG12J058, 2012.*

*Wettlaufer, J. S., and Worster M. G.: Premelting Dynamics, Annu. Rev. Fluid Mech., 38(1), 427–452, doi:10.1146/annurev.fluid.37.061903.175758, 2006.*

---

## Author Comment (AC2) · 8 Sep 2017

*Dear Anonymous Referee #2,*

*We thank you for your valuable feedback. We have reviewed all your comments/suggestions, and have attempted to address each to the best of our ability below. For convenience, your comments are reproduced below, our replies to them are in italics, and revised excerpts from the manuscript are noted between double quotes.*

The manuscript by Osman et al describes a very detailed study on the migration of MSA in ice cores. A crucial finding is correlation between the depth at which migration is observed and the accumulation site, asking for caution when concluding the absence of MSA migration in short cores. Further, the role of a number of physical parameters and processes on MSA migration is discussed in detail, it turns out that accumulation rate is an important one. Last but not least, diffusivity coefficients are suggested based on detailed modelling work on a high resolution ice core. I'm impressed how the study combines expertise in ice core analysis with fundamental physical chemistry. Therefore, it is of paramount interest to a wide scientific audience. The manuscript is very long due to the wealth of information and the careful and precise description of the analysis. It reads very charming, the conclusions are well justified, and assumptions and uncertainties in the analysis are openly mentioned. I refrain from recommending immediate publication, because I need clarification regarding the fundamental aspect in discussing and applying the phase diagrams. These fundamental details are directly linked to the conclusion of the manuscript and one of the questions raised in the introduction "Why should MSA in particular exhibit migrations, while associated soluble impurities and acids do not?".

My concern comes down to the point, that I can't follow how the impurity transport model by Rempel as presented in the manuscript leads to a transport from summer to winter layers via concentration driven diffusion. I can think of two scenarios:

(C1) At T above -20°C, isolated patches of NaCl (winter) and NaCl/MSA (summer) solution form. If temperature increases, volume of the liquid brine increases. At a specific T the two patches might meet. If they do not mix, MSA will diffuse from summer to winter resulting in a constant concentration. On first approximation NaCl might have the same concentration in both patches, so it does not diffuse. This scenario will not build up a new peak at the winter location in the core, but rather smooth the MSA over the whole year.

*As per the model by Rempel et al. (2002) and its linearized form presented in the manuscript, the movement of MSA is not only driven by its own concentration gradient but also by the concentration gradients of other species, such as $Na^+$. This is particularly clear in the linearized model, where MSA migration is shown to be the sum of (i) Fickian diffusion along the $MS^-$ concentration gradient and (ii) an effective advection driven by the vertical variations in $Na^+$ concentration (eq. 13). As shown by the formal development in Section 4.3 of the manuscript (eq. 13 in particular; Page 23, line 4) and illustrated in figure 11 of the manuscript, any local maxima present in a layer where $Na^+$ shows a*

*unique maximum will be carried to the level of the [Na$^+$] maximum. It is an advective effect, fundamentally different from the smoothing effect expected from Fickian diffusion with constant diffusivity.*

*If [Na$^+$] is uniform over the thickness of an annual layer, then the advective effect on MSA migration will vanish, and MS$^-$ will simply diffuse along its own concentration gradient and not show concentration peaks, as correctly pointed out for the Reviewer. However, [Na$^+$] is not observed to be uniform over an annual layer in ice cores (as illustrated in this study as well as in many others), most likely because of the high-frequency (i.e., sub-annual) variability of Na$^+$ deposition. As such, this scenario is currently untested.*

(C2) At T between -75 and -30_C (MSA is still in solution, but NaCl and NaMS are solid). Thus a liquid patch at summer location holding only MSA will form. If that spreads or moves it might meet NaCl crystals. There, crystallisation of NaMS could occur, which will built up a concentration gradient and lead to diffusion of more and more MSA towards the winter layer. This might indeed lead to a complete shift of the MSA peak from summer to winter. But, is the diffusion of MSA in liquid rate determining, or the spreading/movement of the film, or the solution of NaCl, or the precipitation of NaMS? I attached a graph for illustration. It is very likely that I miss an important point here, but may I ask you to clarify the ultimate process in more detail?

*We deeply appreciate your comment, in particular your taking the time to illustrate your point in a graph. As noted in the manuscript (Sect. 2.3), the hypothesis referred to in the comment was indeed initially speculated early on by Mulvaney et al. (1992) in their original study on the migration phenomenon, though these authors lacked empirical constraints on the eutectic temperature of the $CH_3SO_3Na \cdot nH_2O - H_2O$ system. We again agree with the Reviewer's intuition here, and thank him/her for turning us to this point, and particularly so as the comment cued us to a typo in the original document where we aimed to address (albeit less extensively) such a scenario. The typo was corrected, and a more explicit discussion on the matter was added to Sect. 5.1 of the manuscript (Page 27, lines 6-21 and page 28, lines 10-23):*

*(Page 27, Lines 6-21)*

*"In section 2.3, we tested the hypothesis that post-depositional formation of winter [MS$^-$] maxima occurs solely as a result of the precipitation of MS$^-$-salts from their grain boundary solutions in sea-salt rich winter layers (Mulvaney et al., 1992; Wolff et al., 1996; Kreutz et al., 1998, Pasteur and Mulvaney, 2000; Curran et al., 2002). This hypothesis, denoted below as the "Mulvaney model", suggests that MS$^-$ in under-cooled solutions should migrate along its concentration gradient via Fickian diffusion until reaching Na$^+$-rich layers, where crystallization of $CH_3SO_3Na$ removes MS$^-$ from the premelt solution, thereby perpetuating a [MS$^-$] gradient between summer and winter layers in the residual premelt. Importantly, it suggests MSA migration would be inhibited at sites where in situ temperatures are greater than the eutectic temperature of $CH_3SO_3Na \cdot nH_2O - H_2O$ (-29.3°C), since $CH_3SO_3Na$ would not be precipitated from the premelt liquid. However, such an inhibition is not apparent in our data compilation (Sect. 2.3).*

*The RWW model (Sect. 4) is fundamentally different than the Mulvaney model. The RWW model does not represent crystallization and metathetic removal of constituents from the liquid phase (Sect. 2). Rather, in the RWW model, MS$^-$ is implicitly assumed to remain dissolved in the premelt liquid following migration from the summer to winter layers, provided in situ temperatures exceed the eutectic temperature of the binary system $CH_3SO_3Na \cdot nH_2O - H_2O$.*

*(Page 28, Lines 10-23)*

*A currently poorly constrained situation arises for sites characterized by in situ temperatures less than ~ -30˚C and greater than -75˚C (Table 2) In this temperature regime, MSA remains in solution while $Na^+$ is presumably immobile, either as solid state NaCl, $CH_3SO_3Na$, or $Na_2SO_4$ (Table 2). MSA migration as envisioned in the Mulvaney model, but not in the RWW model, may operate under such conditions. On the other hand, the Mulvaney model may not apply should summer concentrations of $Na^+$ be high enough to sequester a large fraction of the $[MS^-]$ as $CH_3SO_3Na$ (s) in summer layers. This sequestration process appears supported by the lack of discernable MSA migration in the subannually-resolved portion (i.e., down to ~ 10.5 m) of the $[MS^-]$ record from South Pole (SP-95), where annual mean SAT is -51˚C (Meyerson et al., 2002). While SP-95 is not considered in our data compilation due to the site's low $\dot{b}$ (0.08 m w.eq. $yr^{-1}$), the lack of clear MSA migration at SP-95 departs from the expected relationship found between $\dot{b}$ and $z_{fo}$ in Antarctica (Sect. 2.1; Fig. 2). This observation leads us to speculate that $MS^-$ at SP-95 may be immobilized in the summer layers through a metathesis reaction with $Na^+$ allocated to the grain boundaries."*

(C3) As a third option, could MSA also be pushed into the gas phase from solution and be transported by gas-phase diffusion? May I ask you to comment on this aspect. (M. H. Kuo, S. G. Moussa and V. F. McNeill, Atmos. Chem. Phys., 2011, 11, 9971–9982.)

*The model presented by Kuo et al. (2011) contrasts loss processes of soluble and volatile impurities by modeling the solubility of the impurities in the solid (ice crystal) water phase and (or) their release to the gas phase during surface melting (i.e., at the grain boundary-vapor interface). We envisage that these processes may be relevant in near-surface (nominally, <1-2 m depth) layers of the polar firn pack, where snow-density is low (effective porosity is high, allowing increased volatile loss through the interconnected pore space of ice grains; Wolff et al., 1996; Domine et al., 2008; Bartels-Rausch et al., 2013) and thermal fluctuations rather extreme (via diurnal, seasonal cycles). These surficial processes thus appear to be useful in predicting the mass conservation of originally-deposited $MS^-$ at some sites. However, as the mechanism of MSA migration explored here appears to be primarily limited to depths deeper than 2 m (Sect. 2), we assume in the manuscript that this volatile post-depositional redistribution plays a negligible role in MSA migration below 2 m. Indeed, empirical evidence by Weller et al. (2004) suggests that, whereas volatile losses of $MS^-$ (as well as $NO_3^-$) can be severe in the upper 1.2-1.4 m of the snow and (or) firn layer at low accumulation sites (<0.10 m w.eq. $yr^{-1}$), such losses (and gaseous-redistribution) are found to be negligible below this depth. These authors attributed the losses in the upper layers jointly to volatile acid formation in acidic (summer) snow layers that can be partially remitted to the atmosphere, a process akin to that described in depth by Kuo et al. (2011). We have included the following text (Page 5, Lines 10-15) making it explicit that we assume no vertical volatile redistribution of MSA:*

*"Post-depositional surficial losses of MSA may occur via gaseous diffusion in the top 1-2 meters of the firn at low accumulation sites (Wagnon et al., 1999, Delmas et al., 2003, Weller et al., 2004). As a result, we exclude records from sites where annual mean accumulation rate is less than 100 kg $m^{-2}$ $yr^{-1}$, and assume that vertical redistribution of MSA via gas-phase diffusion (Kuo et al., 2011) is negligible at all considered sites and depths"*

*(C4)* Second, I would like to read more about the grain boundary network along which migration

of the MSA takes place. A) What is the crystal size, grain boundary density at the position where MSA migration is observed? B) The diffusivity that is discussed is then an effective diffusivity in a porous medium like water/sand or air/snow. When comparing diffusivity between different ice cores or between single crystals and ice cores the grain boundary density (or its volume fraction) needs to be taken into account. I acknowledge that –taken the missing data- this is not possible, but would encourage a more detailed discussion on this issue (F. Dominé, M. R. Albert, T. Huthwelker, H.-W. Jacobi, A. A. Kokhanovsky, M. Lehning, G. Picard and W. R. Simpson, Atmos. Chem. Phys., 2008, 8, 171–208.)

> *Question A) To our knowledge, there is little empirical knowledge on the grain boundary density and crystal size at locations where MSA migration takes place. Thus while we acknowledge the importance of both parameters in acquiring an improved understanding of the migration process, we are severely limited in our ability to investigate this process more fully in this present study, and thus do not discuss further. We have now extended the discussion of Sect. 4.4 to make these points more explicit, namely with respect to constraining the MS⁻ diffusivity (Pg. 26, Lines 21-26; see Question B for denoted text).*

> *Question B) We agree with the Reviewer that the MS⁻ diffusivity that is discussed in our manuscript should be regarded as an effective diffusivity. The revised manuscript includes a paragraph where this point is elaborated (Pages 26-27, Lines 17-29 and 1- 3):*

> *"While the $D_{MS}$ range estimated by a comparison to DIV2010 data is instructive, we note it is not necessarily universal, as diffusivities in polar ice are expected to vary in response to multiple glaciological factors. For example, the experimental results of Kim et al. (2008) show that the diffusion coefficients of ions in under-cooled mixtures are a function of both ionic concentration and temperature. Additionally, physical properties of the firn and ice, including porosity, grain-boundary density, and crystal size, may affect the partitioning of chemical impurities between the liquid premelt and the ice lattice (Dominé et al., 2008; Spaulding et al., 2011), thereby affecting the amount of impurities subjected to anomalous diffusion as well as the interconnectivity of the liquid premelt/vein network. While the RWW model can account for this partitioning (Rempel et al., 2002), the proportions of total MS⁻ and $Na^+$ that are present in liquid form remain poorly constrained (Sakurai et al., 2010). Even at a given site, seasonal and interannual variations in impurity concentrations may lead to down-core changes in $D_{MS}$. Finally, $D_{MS}$ does not take into account whether MS⁻ migration is dominated by diffusion at two-grain boundaries, or at triple junctures and node networks (Wettlaufer and Worster, 2006, Riche et al., 2012). As a result of all these complicating factors, $D_{MS}$, as defined in the RWW model and constrained here, should probably be viewed as an effective diffusivity."*

> *We thank the reviewer pointing us towards the study of Domine et al. (2008), which is now discussed/cited in the revised manuscript, shown above.*

Minor comments

C5) Page 7 line 10 ff: Here I wonder, if the observation of MSA migration at these depths is a matter of time rather than ice density at that depth. Time is mentioned in the intro to this 2.1 but then I miss a discussion or final conclusion on time.

> *We have removed the mention of "time" in the introduction to Section 2.1 (Pg 6, line 15).*

> *We consider the depth of first migration occurrence ($z_{fo}$), as opposed to the timing of first migration occurrence ($t_{fo}$), in Section 2 due to the relatively fewer assumptions and ambiguities associated with $z_{fo}$ as opposed to $t_{fo}$.  For example, $t_{fo}$ could be reasonably defined either as the difference between the*

*surface age and the prescribed age at $z_{fo}$, or as the difference between the age at the threshold migration onset density (Sect. 2.4) and $t_{fo}$. While the latter is preferable in that it suggests the "true" timespan required for MSA migration, it also would require a clear understanding of the migration onset density and corresponding depth, which remains poorly constrained (see expanded discussion in Sect. 2.4). Finally, $z_{fo}$ is impervious to age-depth error, whereas $t_{fo}$ is not. As a result of these limitations, the occurrence of MSA migration is defined in terms of depth, not of time, in our analysis.*

C6) Page 18 line 4 ff: "The RWW model as applied to the binary system containing MS and NA+". This confuses me. The binary system is water-NaMS. Or, do we have a ternary system water-NaCl-MSA?

*We thank the reviewer for pointing out this potentially confusing wording. No, we do not have a ternary system water-NaCl-MSA. In Sect. 4.2.2, we arrive at the conclusion that the binary system being modeled is indeed the $CH_3SO_3Na \cdot nH_2O$-$H_2O$ binary system, though we acknowledge the ambiguity of "binary" prior to this conclusion. We have removed "binary" from the noted sentence (page 18, line 6), and changed all other wordings to explicitly denote the $CH_3SO_3Na \cdot nH_2O$-$H_2O$ binary system when "binary" is indeed implied (e.g., pg. 21 lines 1-2).*

C7) Connected to this: Page 19 line 10: How can an ion have a liquidus curve? The phase diagram is different for each counter ion.

*The parameters $\Gamma_{MS}$ and $\Gamma_{Na}$ may be taken as shorthand notations for $\Gamma_{MS*}$ and $\Gamma_{Na*}$, where \* represents an unknown cationic – anionic pair, respectively. Indeed, exploration of the likely cationic – anionic pair for MS- and Na+, respectively, forms the remainder of 4.2.2's discussion. Text has been added to express this "shorthand" notation explicitly (page 19, lines 17-19):*

*"Knowledge of $\Gamma$ requires knowledge of the dominant precursor (bonded) molecular state(s) of the $MS^-$ and $Na^+$ ions present in the ice (thus, $\Gamma_{MS}$ and $\Gamma_{Na}$ should be viewed as shorthand notations for $\Gamma_{MS*}$ and $\Gamma_{Na*}$, where \* represents some unknown cationic – anionic pair)."*

C8) Page 18 line 18: I suggest to add and discuss the work of Domine on diffusion in single crystals (E. Thibert and F. Dominé, J. Phys. Chem. B, 1998, 102, 4432–4439; F. Dominé and E. Thibert, J. Phys. Chem. B, 1997, 101, 3554–3565.) and discuss the role of grain boundaries with respect to diffusion in porous media in more detail (last point hold throughout the text).

*We thank the Reviewer for pointing us to these studies. In the revised manuscript, we now cite and briefly summarize these two studies (page 18, lines 19-23):*

*"Notably, this estimate is 1-3 orders of magnitude larger than that reported for solid-state diffusion of HCl (Thibert and Dominé, 1997), $HNO_3$ (Thibert and Dominé, 1998), HCHO (Barret et al., 2011), and deuteriorated water (Lu et al., 2009) determined in single ice crystals, despite the molecular radius of MSA greatly exceeding that of each of these species (Roberts et al., 2009)."*

I hope you find these comments helpful and I'm looking forward to your revised manuscript. Please also note the supplement to this comment: https://www.the-cryosphere-discuss.net/tc-2017-84/tc-2017-84-RC2-supplement.pdf

[revised manuscript text omitted]

---

## Editor Comment (EC1) · B. Alexander (Editor) · 12 Sep 2017

Dear Matthew Osman,

Thank you for submitting your responses and revisions. In addition to the reviewer comments that you have already addressed, I have a few minor suggestions detailed below that I think will improve clarity. Please let me know if you have any questions.

Regards, Becky Alexander

Page and line numbers below refer to the revised manuscript.

Page 2 Line 14: Photolysis also results in postdepositional alteration of some chemical species. Perhaps this should be explicitly mentioned in this general statement about

possible processes impacting chemical species in ice cores.

Page 3 Line 9: "lack of wintertime MSA deposition" implies zero MSA deposition flux during winter. Do we really know that it is zero? MSA has an atmospheric lifetime of several days, and thus can be transported over significant distances before deposition. Perhaps replace "lack of" with "relatively low"?

Page 3 line 11: "OH-" should read "OHĔŚ" or "OH radical".

Page 6 lines 19 and 24 have confusing wording. "increasingly shallow depths" is confusing. Maybe replace with a description of the relationship between snow accumulation rate and the depth over which MSA migration occurs. "depths lower" is also confusing. Does "lower" mean above or below a certain depth in this context?

Page 9 line 7: Does this mean that the depth at which movement of MSA stops is deeper, or that MSA moves a larger distance in total?

Page 9 line 8: Can "small values" be replaced with "low concentrations"?

Page 20 line 7: The textbook Seinfeld and Pandis [2006] is probably not the appropriate reference for the Cl:Na molar ratio in seawater.

Page 32 line 6: Can you be more quantitative with the statement "high accumulation and low core-averaged [Na+]" by giving some numbers of range of numbers for "high" and "low"?

---

## Author Comment (AC3) · 27 Sep 2017

B. Alexander (Editor)

beckya@u.washington.edu

*Dear Dr. Alexander,*

*We thank you for your suggested technical corrections, which are addressed below. For convenience, your comments are reproduced below, our replies to them are in italics, and excerpts from the text are between double quotes.*

Dear Matthew Osman,

Thank you for submitting your responses and revisions. In addition to the reviewer comments that you have already addressed, I have a few minor suggestions detailed below that I think will improve clarity. Please let me know if you have any questions.

Regards,
Becky Alexander
Page and line numbers below refer to the revised manuscript.

**Suggested technical corrections**

C1) Page 2 Line 14: Photolysis also results in postdepositional alteration of some chemical species. Perhaps this should be explicitly mentioned in this general statement about possible processes impacting chemical species in ice cores.

*We have now included "photolysis" in the opening paragraph of the manuscript (Pg. 2, Line 14):*

*"Processes acting within the upper firn layer, including wind pumping, diffusion, photolysis, volatility, sublimation, and melt (Wolff et al., 1996), can affect the stability of chemical species… ( )."*

C2) Page 3 Line 9: "lack of wintertime MSA deposition" implies zero MSA deposition flux during winter. Do we really know that it is zero? MSA has an atmospheric lifetime of several days, and thus can be transported over significant distances before deposition. Perhaps replace "lack of" with "relatively low"?

*We agree with this suggestion,and have replaced "lack of" with "relatively low" (Pg. 3, Line 9):*

*"Conversely, the relatively low wintertime MSA deposition may be jointly attributed to…( )"*

C3) Page 3 line 11: "OH-" should read "OHËS´ " or "OH radical".

*We have changed "OH˙" to "OH radical" on Pg., 3, line 12.*

C4) Page 6 lines 19 and 24 have confusing wording. "increasingly shallow depths" is confusing. Maybe replace with a description of the relationship between snow accumulation rate and the depth over which MSA migration occurs. "depths lower" is also confusing. Does "lower" mean above or below a certain depth in this context?

*We have removed "increasingly shallow depths" on Pg. 6, line 19, and have reworded the sentence as follows (Pg. 6):*

*"At low-to-moderate accumulation rate sites ($\dot{b}$ = 0.1 – 0.45 m w. eq. yr$^{-1}$), MSA migration seems to universally occur, with the shallowest reported depths of migration showing a positive relationship with accumulation rate (Table 1 and Fig. 2)."*

*On Pg. 6, Line 24, we have changed "depths lower" to "depths deeper".*

C5) Page 9 line 7: Does this mean that the depth at which movement of MSA stops is deeper, or that MSA moves a larger distance in total?

*On Pg. 9, Line 7, we make the observation that the shallowest observed depth of MSA migration ($z_{fo}$) appears to occur at deeper depths for sites with lower core-averaged $Na^+$ concentrations ($\overline{Na^+}$). This observation does not necessarily relate to the distance traveled by MSA from a summer layer to a winter layer. The sentence referred to on Pg. 9, Line 7 has been adjusted to read as follows:*

*"We find that, as $\overline{Na^+}$ decreases, MSA migration tends to be observed at greater depths in the firn or ice column (Fig. 3)."*

C6) Page 9 line 8: Can "small values" be replaced with "low concentrations"?

*We have changed "small values" with "low concentrations" (Pg. 9, Line 8).*

C7) Page 20 line 7: The textbook Seinfeld and Pandis [2006] is probably not the appropriate reference for the Cl:Na molar ratio in seawater.

*The reference for Seinfeld and Pandis (2006) has been replaced with that of Chesselet et al. (1972; see references below), who similarly adopted a molar ratio of Cl:Na = 1.8 for seawater to study variations in ionic ratios of marine aerosols.*

C8) Page 32 line 6: Can you be more quantitative with the statement "high accumulation and low core-averaged [Na+]" by giving some numbers of range of numbers for "high"and "low"?

*Our results suggest that the timing/depth of MSA migration is a continuous function of site accumulation rate ($\dot{b}$; positive relationship) and core-averaged $[Na^+]$ ($\overline{Na^+}$; negative relationship), indicating that migration should inevitably occur provided i) enough time and (or) depth and ii) requisite thermal conditions (Sect. 5.1). As such, a conscientious decision was made in refraining from providing a specific range of $\dot{b}$ and $\overline{Na^+}$ values for which MSA migration may be deemed "negligible". We provide two primary reasons in support of this decision:*

1. *Studies using [MSA] for paleoclimatic inferences vary in terms of timescales and temporal resolutions that are explored/achieved (Table 1), indicating that values of $\dot{b}$ and $\overline{Na^+}$ which are relevant will similarly vary from study to study.*
2. *Following reason (1), irrespective of the site-specific values of $\dot{b}$ and $\overline{Na^+}$, the precise timing of MSA migration still remains broadly unconstrained, given in particular the significant uncertainty in the grain-boundary diffusion coefficient of MSA (Sect's. 4.4; see also Fig. 14).*

*Thus, providing quantitative constraints on "high accumulation and low core-averaged $[Na^+]$" may be subjective and (or) potentially misleading. Rather, we believe the associated quantitative results are well encapsulated within Figures 14 and 15. As such, no changes were made to Pg. 32, Line 6.*

**References**

Chesselet, R., Morelli, J., and Buat-Menard, P.: Variations in ionic ratios between reference sea water and marine aerosols, J. Geophys. Res., 77(27), 5116–5131, doi:10.1029/JC077i027p05116, 1972.